

# Low cloud reduction within the smoky marine boundary layer and the diurnal cycle

Jianhao Zhang[1] and Paquita Zuidema[1]

[1]Rosenstiel School of Marine and Atmospheric Science, University of Miami, 4600 Rickenbacker Cswy, Miami, FL, 33149 USA

**Correspondence:** Jianhao Zhang (jzhang@miami.edu) and Paquita Zuidema (pzuidema@miami.edu)

**Abstract.**

Previous observational studies of the southeast Atlantic emphasize an increase in the stratocumulus cloud cover when shortwave-absorbing aerosols are present in the free-troposphere. Recent field measurements at Ascension Island (8°S, 14.5°W) reveal that smoke is also often present in the marine boundary layer, most evident in August when the smoke is highly absorbing

of sunlight, the boundary layer is deeper, the cloud-top inversion is weaker, and a climatologically lower cloud fraction eases penetration of the sunlight to the surface, compared to later months. In these conditions, the low cloud cover decreases further with enhanced smoke loadings, reflecting a boundary layer semi-direct effect that is a positive feedback. The low cloud cover reduction is particularly pronounced in the afternoon, although the cloud liquid water path is more strongly reduced at night. The daily-mean surface-based mixed layer is warmer by approximately 0.5 K when more smoke is present in the boundary

layer, with a warming peak in the late afternoon when the cloud cover reduction is largest. After sunset, sub-cloud moisture accumulates throughout the night, increasing the moisture stratification with the cloud layer. This increase in boundary layer decoupling is consistent with reduced turbulence. A new observation is that in the sunlit morning hours, the smokier boundary layer deepens by approximately 200 m, and both the liquid water paths and cloud top heights increase. We speculate this reflects radiatively-induced vertical ascent originating from within a well-mixed smoke-filled sub-cloud layer. Overall, the reduction in

daytime low cloud decreases the top-of-atmosphere all-sky albedo, despite an increase in the top-of-atmosphere direct aerosol radiation of ∼6.5 W m$^{-2}$ between the (most-least) smoky tercile composites. A convolving meteorological influence is also apparent near the cloud top, in that, although the free troposphere is also often smoky in August, the associated above-cloud potential temperatures are often cooler, rather than warmer, and better-mixed. The cooling weakens the inversion beyond that expected from the warming of the boundary layer and further encourages entrainment of more smoke into the already smoky

boundary layer, increasing the longevity of the boundary layer smoke events. The free-tropospheric winds are also typically stronger and more easterly. More smoke appears to settle into the sub-cloud layer during the day than at night when it is smoky, speculated to reflect a deeper daytime sub-cloud layer facilitating entrainment, when the nighttime stratification does not.



## 1  Introduction

Shortwave-absorbing aerosols above the southeast Atlantic overlay and mix in with one of the earth's largest stratocumulus deck from July through October. Many studies highlight the presence and radiative impact of the absorbing aerosol in the free troposphere (Waquet et al., 2013; Peers et al., 2015; Das et al., 2017; Sayer et al., 2019; Peers et al., 2019; Deaconu et al., 2019), and indeed recent aircraft measurements confirm this for the month of September (LeBlanc et al., 2019; Cochrane et al., 2019). A stratocumulus thickening occurs beneath the layers of the biomass-burning aerosols (Wilcox, 2010), along with increases in the stabilization of the lower free troposphere, cloud cover, and top-of-atmosphere all-sky albedo (Adebiyi et al., 2015). The strengthening of the low cloud deck has been attributed to the above-cloud aerosol shortwave absorption, strengthening the capping inversion and reducing entrainment (Johnson et al., 2004; Gordon et al., 2018), mediated perhaps through a coupling to meteorology and/or combined with aerosol indirect effects (Fuchs et al., 2018; Adebiyi and Zuidema, 2018).

The process by which the clouds adjust to the presence of the absorbing aerosols depends critically on the relative location of the aerosol to the low cloud deck (Johnson et al., 2004; Koch and Del Genio, 2010). Aerosol-cloud microphysical interactions, indicating that smoke can also be present in the boundary layer, have also been observationally documented (Costantino and Bréon, 2013; Painemal et al., 2014; Grosvenor et al., 2018; Diamond et al., 2018) and modeled (Yamaguchi et al., 2015; Zhou et al., 2017; Lu et al., 2018) for the southeast Atlantic. It has been difficult to infer the presence of boundary-layer absorbing aerosols directly, however. In addition, to date, no observational studies (that we are aware of) have documented how clouds adapt radiatively to the presence of smoke within the southeast Atlantic boundary layer. Priori to 2016, only one research aircraft campaign had documented the presence of smoke in the boundary layer of the remote SE Atlantic (Haywood et al., 2003). This marine boundary layer remained unsampled until 2016, when the Department of Energy Atmospheric Radiation Measurement program deployed its Mobile Facility 1 (AMF1; Miller et al., 2016) to the windward side of Ascension Island as part of the Layered Atlantic Smoke Interactions with Clouds (LASIC) field campaign (Zuidema et al., 2015), as one of several new campaigns focused on the southeast Atlantic (Zuidema et al., 2016b). Ascension Island (8°S and 14.5°W) is midway within the South Atlantic basin, about 2000 kilometers to the west of continental Africa, and underneath the main outflow region of the biomass-burning-aerosol plumes from continental African fires during June through October (Adebiyi et al., 2015). Early results indicated the frequent presence of biomass-burning aerosols near the surface, with maxima in black carbon mass concentrations in August (Fig. 1), particularly pronounced in 2016.

Studies focusing on the smoke-filled boundary layers of other marine regions have shown that shortwave-absorbing aerosols can decrease the relative humidity by raising the temperature, encouraging the reduction in low-cloud cover (e.g. Hansen et al., 1997; Ackerman et al., 2000; Lohmann and Feichter, 2005). Two recent studies (Pistone et al., 2016; Wilcox et al., 2016), from the CARDEX campaign (Cloud, Aerosol, Radiative forcing, Dynamics EXperiment) based in the northern Maldives during 2012, found that polluted conditions coincide with reduced turbulence, with less entrainment at the top of the sub-cloud layer resulting in a shallower surface mixed layer with enhanced humidity. In contrast to the suppressed polluted shallow marine cumuli examined within Ackerman et al. (2000), Pistone et al. (2016), and Wilcox et al. (2016), the low clouds at the location of Ascension Island occupy deeper boundary layers ranging from 1.2 to 2 km (Fig. 2). The clouds often occupy two layers,



one with cumuli bases rooted at the lifting condensation level at ∼700 m, and the other reflecting stratiform outflow capped by the trade-wind inversion at ∼1650 m (Fig. 2). The monthly-mean profiles also indicate a prevalent decoupling between the upper and lower portions of the boundary layer. The vertical profiles reflect Ascension's location at the northwestern edge of the stratocumulus deck, a deck whose coverage peaks in September-October (Zuidema et al., 2016a). At Ascension Island, the warm cloud cover ranges from 10% to 70%, with a broad September-November maximum and clear morning to afternoon difference (Fig. 3).

The August maxima in the near-surface refractory black carbon mass concentrations motivate this study's focus on documenting that month's low cloud response. The variability in the smoke loading is similar between the two years (Fig. 1), suggesting the two August months, from two different years, can be combined. Smoke loadings can remain elevated for over a week in August, punctuated by shorter time periods with low smoke loadings. The variability occupies shorter time scales in June and July, while in September the boundary layer smoke loadings decrease dramatically. Single scattering albedos range from 0.78 to 0.83 (Zuidema et al., 2018), values that are lower (more absorbing) than documented for the free troposphere (Haywood et al., 2003; Pistone et al., 2019; Cochrane et al., 2019), suggesting a robust radiative response is likely.

This radiative response is primarily assessed through the diurnal cycle. This takes advantage of the systematic sampling across the diurnal cycle that is available from the LASIC site, and the idea that the diurnal cycle may be able to provide additional insights into the boundary layer semi-direct effect not available to aircraft campaigns with daytime-only sampling strategies. The diurnal time scale is faster than that of meteorology, holding out hope that the effects of shortwave absorption, which is only active during the day, can be distinguished from those associated with meteorology.

The two separate cloud layers at Ascension are indicative of cumulus-under-stratocumulus, which can have distinct diurnal cycles. Surface observers report that stratocumulus is the dominant cloud type during June, July and August, with a broad maximum in cloud cover around dawn, reflecting accumulated nighttime radiative cooling (e.g., Rozendaal et al., 1995; Eastman and Warren, 2014; Painemal et al., 2015; Seethala et al., 2018). Surface-forced cumuli clouds, in contrast, have a cloud fraction maximum in the late afternoon, near sunset (Eastman and Warren, 2014, Fig. 4 and 6), with a minimum at sunrise. Similar diurnal cycles are observed during the summer months in other eastern subtropical oceans, e.g. in the NE Atlantic (Ciesielski et al., 2001) and the NE Pacific (Fig. 4 in Klein et al., 1995). Other studies emphasize that the diurnal cycle becomes more pronounced as low cloud becomes more broken downstream of the main stratocumulus deck, (Rozendaal et al., 1995; Burleyson and Yuter, 2015), as is the case for Ascension Island, suggesting the diurnal cycle response documented at Ascension to the presence of smoke is likely to be robust.

One benefit for this study is that, in August of 2016, eight radiosondes were launched daily, an increase from the campaign-norm of four times daily. These form the first measurements of the thermodynamic structure diurnal cycle at this location. Further campaign site information, the dataset, an overview of the August time periods, and methods are described in Section 2. The diurnal cycle in cloud properties as a function of the smoke loading is put forth in Section 3, with thermodynamical and microphysical explanations explored in Section 4. Examples from individual days illuminating features of the general diurnal cycle are shown in Section 5. The impact on the top-of-atmosphere all-sky albedo is discussed in Section 6. Free-tropospheric



conditions and meteorological influences are discussed in Section 7. Positive feedback mechanisms are discussed in Section 8, followed by synopsis and discussion of the diurnal cycle in the last section.

## 2    Data, overview and compositing approach

The highest point on the volcanically-produced island is the 859 m peak of Green Mountain, of which the AMF1 site is located
on its windward flank at 340 m above sea level. Orographic lifting clearly induces additional cloud at the lifting condensation level (LCL; ∼700 m at Ascension during August), obvious within the vertical distribution of ceilometer-detected cloud base heights at the AMF1 site compared to those from the airport located four km to the west, away from the volcano (Fig. 4 dashed curves), and within satellite visible imagery under suppressed conditions. The diurnal cycle in low cloud properties is therefore primarily evaluated using retrievals from a geostationary satellite and the measurements available from the airport.
The aerosol extinction retrievals from the micropulse lidar at the AMF1 are also affected by attenuation from the lower-lying cloud. In contrast, the AMF1 site precipitation measurements are thought to be more representative of the larger region. This is subjectively based on satellite visible imagery that does not indicate an obvious impact from the mountain peak on the deeper boundary layers associated with the larger precipitating systems. A quantitative assessment might be possible using scanning radar measurements, but is not provided here. The in-situ aerosol measurements, in contrast to the cloud measurements, benefit
from the AMF1 site's location. The site, upon a hard-packed volcanic rock surface with a thin soil, is away from local sources of pollution, and the elevation above sea level reduces the contribution from sea spray (no dedicated sea salt measurements are available). The persistent southeasterly boundary layer winds encourage aerosol measurements that are representative of the open ocean's atmospheric boundary layer, an assumption that will be assessed at a future date using measurements from the UK CLARIFY (Cloud Aerosol Radiation Interactions and Forcing - Year 2017) aircraft campaign, based on Ascension in
August-September of 2017 (Zuidema et al., 2016b).

### 2.1    Ascension island measurements

The primary measure of smoke is refractory black carbon mass concentrations derived from a single particle soot photometer (Fig. 1) and secondarily carbon monoxide (CO) concentrations. Aerosol vertical structure information comes from extinction profiles derived from the 532 nm wavelength micropulse lidar (MPL) following Delgadillo et al. (2018). The MPL signal will
be fully attenuated by a cloud with optical depth greater than one (Delgadillo et al., 2018), and an explained variance ($R^2$) exceeding 0.5 is required between the normalized relative backscatter signals to a calculated molecular scattering profile above the aerosol layer, indicating that the lidar was able to penetrate the aerosol layer. A new overlap function was applied after the LASIC campaign, producing reasonable agreement between newly-derived extinction profiles and those from a Raman lidar at the Southern Great Plains DOE site (Paytsar Muradyan, pers. comm.). One-hour averaged cloud condensation nuclei
(CCN) concentrations at 0.4% supersaturation are derived from column A of a dual-chamber cloud condensation nuclei particle counter at the AMF1 site.



Vaisala RS-92 radiosondes were launched eight times daily in August 2016 and four times daily in August 2017 at the airport, allowing profiling of the complete boundary layer and yielding 357 successful launches for the two Augusts. The lowest 100 meters of the radiosonde profiles appeared to reflect the island's surface heating effect, perhaps through their enclosure, and radiosonde profiles lower than 100 m are disregarded. This surface heating effect is most pronounced when the

sun is overhead, i.e. the first hundred meters are warmer than the diurnal mean by ∼1.5 K around noon, decreasing rapidly upwards (not shown). The profiles were further re-gridded to a common ten-meter vertical spacing from the surface to five km. The lifting condensation level is calculated using the temperature and relative humidity of an air parcel originating from 1000 hPa (see Appendix). The cloud-top inversion bases correspond to the maximum height of relative humidities greater than 75 %, and the inversion tops to the local maximum in the saturated equilibrium potential temperature, following Yin and Albrecht

(2000). The depth of the inversion layer is restricted to 500 meter. Visual inspections of the inversion bases and tops are applied as a sanity check when calculating the inversion strengths from the difference in potential temperature between the inversion base and top.

Cloud base heights are measured by the surface-based ceilometers at both the AMF1 site and the airport site, with only the lowest cloud bases reported within each 15 seconds used. One-hour total cloud frequency is derived from the ceilometer

by calculating the ratio of number of samples when cloud bases are detected to the number of total samples within a 1-hour window. Cloud/precipitation frequency altitude distributions are derived from the Ka-band vertical-pointing cloud radar (KAZR) using a threshold of -35 dBZ. The KAZR reflectivities are biased high by 4-6 dB compared to that from the scanning K-band radar, increasing for weaker signals (Brad Isom, pers. comm.). The scanning radar is the more accurate of the two radars, as it was calibrated regularly using a reflector as part of its scanning pattern; the offset is not accounted for here. Only

the cloud liquid water path retrievals from the microwave radiometers at the airport were used. These are physically retrieved every minute from the 2-channel microwave radiometer (23.8 GHz and 31.4 GHz) using the standard ARM operational retrieval algorithm (Turner et al., 2007). Due to a hardware malfunction of the microwave radiometer during August 2016, physical LWPs are instead retrieved from a separate microwave radiometer profiler, also at the airport, for August of 2017. Surface precipitation measurements at every minute are obtained from a disdrometer at the AMF1 site. A one-hour rain frequency is

derived from the disdrometer by calculating the ratio of number of samples with rain rates greater than zero to the number of total samples within a 1-hour window.

The relevant datasets, their uncertainties if known, instrument source and comments are summarized in Table 1, and a full list of the available instrumentation for LASIC can be found in (Zuidema et al., 2015). Further detail on the radiosondes' quality control and selection is provided in the Appendix. All times are reported in local solar time, which is approximately one hour

earlier than the UTC time.

## 2.2   Satellite low-cloud properties

Hourly areal-mean cloud fractions and effective cloud top heights over a four by four degree domain latitudinally centered on Ascension but slightly to the island's east (6°S to 10°S and 15°W to 11°W, done to capture the clouds most representative of the island), are retrieved from radiances measured by the Spinning Enhanced Visible and Infrared Imager (SEVIRI) onboard



the geostationary Meteosat10 satellite using the Visible Infrared Solar-Infrared Split Window Technique (VISST; Minnis et al., 2008). The low-cloud fractions are estimated from the pixel-level products at three km resolution as the ratio of liquid-only water+suspected water pixels to the total retrieved pixels within an one-hour time period. We note occasional misclassifications of near-surface smoke as cloudy pixels when solar zenith angles are low (i.e. sunrise and sunset) by the VISST algorithm,

resulting in apparent water clouds with effective heights lower than 500 m. These "super low clouds" were excluded from the analysis.

The smoke radiative effect is evaluated using daily all-sky top-of-atmosphere albedos and low-cloud fractions available from the Clouds and the Earth's Radiant Energy System (CERES) one-degree Synoptic (SYN1deg) products, from both Aqua and Terra satellite platforms. For context, ten years of monthly low-cloud fractions are also obtained from level-3 one-degree

gridded datasets derived from Moderate Resolution Imaging Spectroradiometer (MODIS) data onboard of Terra and Aqua satellites. Both monthly-mean MODIS and daily-mean CERES products are areal-averaged over the same four by four degree domain. In general, CERES retrievals of low cloud fraction are slightly lower (∼5%) than those from SEVIRI (Fig. 5).

### 2.3   August overview

An overview of the two August time series indicate remarkably similar conditions within the Augusts of the two years, lending

robustness to the analysis (Fig. 5). The carbon monoxide (CO) and rBC measurements indicate that smoke was almost always present near the surface. The exception is 21-23 August of 2017, when CO concentrations are close to the background value and rBC mass concentrations are within the instrument sensitivity. Two pronounced multi-day smoke events are apparent during each of the two months (Fig. 5), with one during mid-august, persisting just more than a week, and the other one at the end of the month, lasting for just a little shy of a week. The 12-16 August 2016 period is the smokiest of the entire campaign,

with rBC mass concentration reaching 1700 ng m$^{-3}$, analyzed further in Zuidema et al. (2018). The daily areal-mean low cloud fraction clearly reduces during this time (Fig. 5, upper panel), while the cloud cover reduction is less dramatic in August 2017 (Fig. 5, bottom panel). Precipitation is also significantly less frequent and less intense during the smoke events (Fig. 5 green curve), either because the liquid water path is reduced, or because increased cloud droplet number concentrations reduce the precipitation susceptibility at a given liquid water path.

### 2.4   Compositing of smoke conditions

Conditional compositing classifies more- and less- smoky conditions based on the refractory black carbon mass concentrations. A daily-mean threshold of 500 ng m$^{-3}$ establishes "more smoky" conditions, and less than 100 ng m$^{-3}$ establishes "less smoky" conditions. These values are approximately based on the tercile values from the August distributions of one-minute rBC mass concentration (Fig. 6). Days with strongly varying rBC mass concentration, indicative of a synoptic change, are not included

by constraining the allowed daily standard deviations (50 and 120 ng m$^{-3}$ for less and more smoky days). This selection yields 13 more smoky days (7 from 2016 and 6 from 2017), and 13 less smoky days (6 from 2016 and 7 from 2017).



## 3   The cloud diurnal cycle as a function of the smoke loading

The domain-averaged SEVIRI total low cloud fractions for August over Ascension, including both stratocumulus and cumulus, peak at ∼2 hours after sunrise, with a minimum at ∼2 hours before sunset (Fig. 7). Although broadly similar to that of other eastern subtropical stratocumulus regions, the cloud cover maximum is slightly delayed, extending into the sunlit hours - a

post-sunrise maximum that is apparent in other cloud properties as well, discussed below. A boundary-layer semi-direct effect is clear: when more smoke is present in the boundary layer, the diurnal-mean low-cloud cover decreases. Of further impact for the diurnally-averaged albedo is that the amplitude of the diurnal cycle also increases because of an enhanced afternoon reduction in cloud cover, particularly in August of 2016.

Interestingly, the changes in the diurnal cycle of the cloud liquid water paths (LWPs; Fig. 8 a and b) do not fully conform

to that in cloud cover. The 3-hour averaged cloud liquid water paths are constructed from one-minute retrievals, which will sample a distance of approximately 400 m at a representative wind speed of 7 m/s, and corresponds more to a measure of cloud thickness than cloud fraction. The two years are shown separately as they are not retrieved from the same instrument, and the August-mean cloud amount is less in 2016 than 2017 (Fig 3). The LWP is clearly less in the afternoon under smokier conditions in 2016, though not so in 2017. A more consistent feature between the two years is the reduction in LWP apparent

after midnight in smokier conditions. Also consistent is an atypical increase in LWP that occurs mid-morning, after the cloud cover has begun to decline.

The diurnal cycle in the three-hour frequency and intensity of rain detected at the surface (Fig. 8 c and d) indicates a nighttime maximum that preceeds the morning maximum in cloud cover. The rain rate diurnal cycle composite follows the rain frequency composite well, with both showing a maxima before dawn and a secondary maximum in the late afternoon. Sea

surface temperature measurements made at a buoy located at 10°S, 10°W confirm a weak afternoon warming (∼0.05 K) of the ocean surface (not shown). The presence of more smoke reduces how frequently rain reaches the surface, although a slight late-night maximum and secondary late-afternoon maximum is still present. The cloud condensation nuclei concentrations (CCN) at 0.4% supersaturation increase to daily-mean values of 790 cm$^{-3}$ for more smoky conditions, compared to 114 cm$^{-3}$ for less smoky conditions. The elevated CCNs will suppress the likelihood of raindrop formation, and appear to do so even

when LWP increases in the mid-morning.

Vertically-resolved cloud fractions derived from the Ka-band zenith pointing cloud radar and aggregated every six hours, support the LWP measurements. Clouds are fewer and may be less vertically-developed in smokier boundary layers (Fig. 9). The more surprising documentation is that the cloud tops are higher in the morning (06 – 12 LST) under smokier conditions, consistent with the higher LWPs between 08 – 10 LST (Fig. 8).

Satellite measurements of cloud top heights as a function of low-cloud fractions and BL smoke loadings confirm the island-based observation. The cloud heights are typically lower under smokier conditions, when they correlate more strongly with lower cloud fractions (Fig. 10). These reach cloud top heights of approximately one km at the lowest cloud fractions, indicating loss of the upper stratiform cloud layer. The exception is in the early morning. Almost all 08 LST samples (Fig. 10 green





crosses) have a cloud effective height higher than 1.5 km, consistent with the higher cloud tops observed at that time by the surface-based radar (Fig. 9).

## 4 Explanations for the altered cloud diurnal cycle

### 4.1 thermodynamic

5   The radiosonde thermodynamic profiles (Fig. 11) indicate boundary layer depths that can exceed the cloud top heights documented in Fig. 10. Furthermore, the mean daytime profiles for the "more smoky" composite in Figure 11 indicate the boundary layer is deeper, by approximately 200 meters, than that of the "less smoky" composite. The deepening of the boundary layer is more well-defined in August 2016, but the mean deepening occurs independently in both years, indicating a common physical cause is likely as opposed to a sampling bias. This deepening contrasts with the decrease in the cloud top heights under smokier conditions (Fig. 10), indicating that the upper cloud layer dissipates under smokier conditions.

The individual profiles of potential temperature ($\theta$), relative humidity ($RH$), and water vapor mixing ratio ($q_v$) are highly variable (Fig. 11). Individual trade-wind inversion heights vary from 1.5 to 2.2 km, with inversions that are sharper than that within the mean profile. A surface-based layer with well-mixed potential temperatures and water vapor mixing ratio ($q_v$) is often present, extending to about 500 meters, identified from here on as "$ML_{SB}$". The decoupled boundary layer structure is evident in the stratification of the water vapor mixing ratio profiles, in which the $ML_{SB}$ $q_v$ exceeds that within the cloud layer above. Mean August $ML_{SB}$ $\theta$ and $q_v$ values of ~296 K, and ~13 g kg$^{-1}$, respectively, are slightly cooler and moister than those within the upper cloud layer, where the mean $\theta$ increases to ~299 K near the cloud top inversion and $q_v$ decreases from around 12 g kg$^{-1}$ just above the LCL to about 10 g kg$^{-1}$ near the cloud top, reflecting either entrainment or conversion to cloud water. The free troposphere above two km is clearly drier and warmer than the boundary layer, with a mean $RH$ and $\theta$ of ~25% and 20   ~311 K, respectively (Fig. 11).

Diurnally-resolved mean thermodynamic profiles indicate that the deepening of the smokier boundary layers occurs during the morning, followed by more significant shoaling during the night (Fig. 12a). The potential temperature of the smokier composite is the most warm near the surface in the mid-afternoon (14 LST), with the warmth moving upward in the late afternoon (17 LST). This translates into a reduced $RH$ throughout the entire lower kilometer (Fig. 12b and d) during the 25   afternoon, consistent with the strongly reduced afternoon cloud cover (Fig. 10). The lowest 400 m become more well-mixed after sunset, when shortwave absorption no longer occurs, and the near-surface layer equilibrates to the ocean's temperature. Slight stable layers appear within the nighttime smokier $\theta$ profiles at approximately 450 m at 20 and 2 LST, with cooler temperatures below. These discourage the upward fluxing of moisture, consistent with the lower nighttime LWPs of Fig. 8. The nighttime $q_v$ increases in the lower 500 m in the smokier composite relative to the "less smoky" composite (Fig. 12c), so that, 30   combined with the warmer temperatures, the nighttime $RH$ remains similar regardless of the smoke loading. By 5 LST, the lowest 500 m of the smokier profile is well-mixed. This pre-conditions the boundary layer air for the vertical ascent needed to increase the moisture and humidity above 500 m in the 8 and 11 LST profiles, with a well-mixed potential temperature through the lowest km. By 11 LST a temperature stratification is again apparent in the lowest km. The warming is greater near the





surface than higher in the boundary layer, which suggests that the absorbing aerosol may be preferentially located near the surface.

The "more smoky" $ML_{SB}$ is warmer by approximately 0.5 K, and more moist by $\sim$1 g kg$^{-1}$ (Fig. 11, c and f), compared to the "less smoky" $ML_{SB}$, so that the mean $RH$ does not vary systematically with the smoke loading (Fig. 11, b and e). The $ML_{SB}$

$q_v$ and $RH$ is typically lower during the day and higher in the night time. $ML_{SB}$-mean thermodynamic quantities, calculated as the mean of the 200–400 m layer, clarify the diurnal cycle of the sub-cloud layer (Fig. 13). The smokier $ML_{SB}$ mean $\theta$ peaks around 14 LST, with a minimum around midnight regardless of the smoke loading, but the afternoon warming extends longer into the afternoon as the smoke loading increases. The smokier $ML_{SB}$ is more moist during the night, by 0.25-0.5 g kg$^{-1}$, than its less smoky counterpart. The sub-cloud $RH$ decreases during the day for higher smoke loadings. At night the sub-cloud $q_v$

and $RH$ track each other, with both peaking around 20 LST for light smoke loadings, and several hours later, around 2 am LST, for heavier smoke loadings. This peak buildup in the sub-cloud moisture preceeds the 4 am LST peak in rain frequency. The sub-cloud $q_v$ and $RH$ decrease after midnight, reaching their diurnal minima at 8am LST, or 3 hour before the diurnal LWP maximum, indicating a similar time lag between the ventilation of the boundary layer and the cloud development.

The diurnal cycle of the lifting condensation level follows the variations in the $ML_{SB}$ RH, such that LCL rises during the day,

reaching a maximum height of 700-800 m around 14 – 16 LST, then collapsing around sunset, and leveling off before midnight (Fig. 13d). Such a diurnal cycle is typical of oceanic stratocumulus, with Hignett (1991) connecting the higher mid-afternoon LCL to a suppression in turbulence. This diurnal cycle becomes more exaggerated under smokier conditions, apparent in a further afternoon lifting of the LCL. During this time, the tenth percentile of the lowest ceilometer-inferred $z_{cb}$ compares well against the LCLs (Fig. 13d), indicating that when clouds do form, they are surface-driven (consistent with the weaker

afternoon precipitation maximum). During the night, $z_{cb}$ exceeds LCL by $\sim$400 m after midnight, indicating that a nighttime recoupling of the boundary layer rarely occurs, in contrast for more well-mixed oceanic stratocumulus (Hignett, 1991; Zuidema et al., 2005). The nighttime lowest detected cloud base can be above 900 m, and is likely a stratiform layer reforming through radiative cooling closer to the inversion base, at the same time that moisture accumulating near the surface. Only in the mid-morning, when increasing LWPs indicate vertical cloud development (Fig. 8a and b), is coupling evident through this metric

(Jones et al., 2011) under higher smoke loadings.

## 4.2 microphysical

The reduction in precipitation frequency under smokier conditions, despite the increase in LWP during the morning, also indicates a microphysical contribution to the morning BL deepening. The cloud condensation nuclei (CCN) concentrations at 0.4% supersaturation attain a mean values of 790 cm$^{-3}$ for the "more smoky" composite, compared to 114 cm$^{-3}$ for the "less smoky"

composite. Non-precipitating shallow convection deepens more than precipitating clouds in equilibrium conditions (Stevens, 2007; Seifert, 2008), with the evaporation of small droplets above the base of the trade-wind inversion from overshooting cumulus turrets pre-conditioning the environment for further cloud development. Thus, the suppression of precipitation also contributes to the morning deepening of the boundary layer.



## 4.3 Diurnal redistribution of aerosol

The vertical distribution of the smoke-absorbing aerosol will affect the radiative heating profile, with a "bottom-heavy' aerosol profile also placing the radiative heating near the surface. Diurnal cycle information on the near-surface rBC mass concentrations, for the "more" and "less" smoky composites, are interrogated for potential insights into their entrainment process from above. Days with daily rBC variation less than 120 ng m$^{-3}$ and 50 ng m$^{-3}$, respectively, are selected from the "more" and "less" smoky days, and detrended. Days with strong short-term entrainment episodes are also excluded. This yield a total of 7 "more smoky" days and 10 "less smoky" days. When the smoke loading is less, the near-surface rBC mass concentrations are highest during the night, with a maximum between midnight-4 LST, and a broad minimum in the late afternoon (Fig. 14). This is consistent with more entrainment when cloud LWP is a maximum, or slightly afterwards, into a well-mixed boundary layer. Under smokier conditions, the near-surface rBC values accumulate after mid-night. This is consistent with concentration within a shallower sub-cloud layer (Fig. 13d). A clear dip in mid-morning of the smokier diurnal cycle is plausibly consistent with a ventilation of moisture and smoke into the cloud layer. Overall, during a typical day, smoke settles towards the surface under smokier conditions, and is advected upwards when the surface layer is less smoky (Fig. 14). The former is consistent with a more mature smoke event, in which smoke slowly moves downwards in the mean. A weaker trade-wind inversion, because of a lack of clouds, may encourage more entrainment from above, with less distance to travel to reach a deeper sub-cloud layer. We only speculate explanations for the diurnal cycle shown in Fig. 14, with a deeper understanding requiring integration with the aircraft data acquired during the CLARIFY experiment.

## 5 Examples

Examples from two days illuminate features of the general diurnal cycle for a smokier boundary layer. Visible imagery from two smoky August days in 2017 (12 and 20 August, with respective daily-mean rBC concentrations of 663 and 346 ng m$^{-3}$), indicate cumulus clusters in the morning, also evident in the cloud radar reflectivity time series (Fig. 15). The cloud radar reflectivities indicate the thinness of the upper-level cloud, at times detected by the ceilometer but not by the reflectivity threshold of -35 dBZ. By 1430 LST, the upper-level stratiform cloud is gone, with only scattered cumuliform clouds remaining. The more developed cumuli clouds occur near or after sunrise on both days, with heavier precipitation reaching the ground on August 20, the less smoky of the two examples. In both cases, the cloud base rises during the day, with more shallow cumuli reappearing after 1400 LST and recurring throughout the night.

The mid-morning cumuli clouds development and BL deepening are well captured by the radiosonde profiles. On August 12, the BL deepens by about 350 m, from ~1350 m at 5 LST to ~1700 m at 11 LST, then remains at the same height until 17 LST, and collapses back to ~1300 m near midnight (Fig. 16a). The morning deepening of the BL is also visible in the radar reflectivities as indicated by the overshooting cumuli cloud tops at 6 LST and 9 LST, with light precipitation reaching the ground (Fig. 15a). A weaker transition layer at the top of the ML$_{SB}$ (~700 m) is evident in the 5 LST sounding, suggesting a decoupled BL, while by 11 LST, the BL $\theta$ is better mixed. The daytime ML$_{SB}$ layer warms by almost 1 K or more.





The $q_v$ profiles from August 12 are highly stratified at 5 LST (Fig. 16a), consistent with a more decoupled nighttime BL aided by the smoke induced suppression of turbulence. This strong stratification of moisture is gone by 11 LST, consistent with moisture ventilated upwards by the morning convection, postulated to be aided by the near-surface smoke radiative heating. The strong stratification in moisture may also help explain why the mid-morning cumuli development was so pronounced, and

why the BL deepening on this day exceeded the August-mean value of $\sim$200 m (Fig. 11 and 12). The cloud base rise evident within the radar time series from noon to 14 LST (Fig. 15a) is consistent with the clear reduction in the $ML_{SB}$ relative humidity (Fig. 16a).

On August 20, when the near-surface smoke loading almost half that of August 12, the same features are present, such as a morning BL deepening, daytime $ML_{SB}$ warming, nighttime decoupling and moisture stratification, and afternoon drying of

$ML_{SB}$ (Fig. 16b). The differences are that the boundary layer was deeper overall, and the BL deepening of 200 m is closer to the average. The daytime $ML_{SB}$ warming is more concentrated closer to the surface, and the moisture stratification at 5 LST is weaker than on August 12. We speculate that with less smoke within a deeper boundary layer, radiative warming by the absorbing smoke does not generate as much buoyancy, so that the cumuli cloud tops do not overshoot as much as on August 12. Another difference is that more rain drops are able to reach the ground on August 20 compared to August 12, consistent

with lower CCN concentrations.

## 6   Top-of-Atmosphere Albedo adjustment

The question arises as to whether the increased aerosol loading, or the cloud reduction in response to the presence of the increased smoke, dominates the overall radiative impact. We discuss a simple estimation of the aerosol radiative impact for a certain cloud fraction (direct aerosol radiative effect) over Ascension, using areal-mean daily CERES all-sky albedo and

low-cloud fraction from a 4°by 4°domain centered slightly to the east of Ascension, as well as surface rBC measurements (Fig. 17). As shown, the TOA all-sky albedo of the smoky samples are less in the mean than those from less smoky days (Fig. 17, histogram in the bottom right corner), indicating an overall decrease in TOA all-sky albedo as the BL smoke loading increases. The linear fits indicate a good correlation between low-cloud fractions and TOA all-sky albedos, with the correlation coefficients being 0.79 and 0.93 for more smoky and less smoky conditions, respectively. Although the high correlation between

low-cloud fractions and TOA all-sky albedos suggest that the TOA all-sky albedo in the remote SE Atlantic is mostly dominated by the changes in low-cloud fraction, the difference between the two fitting lines provide an estimate of the aerosol direct radiative effect, in which smoke cools the atmosphere by reflecting more sunlight back to space for a given cloud fraction.

We further estimate the aerosol radiative impact at two low-cloud fractions by taking the difference between the two fitting lines at cloud fractions of 0.4 and 0.65. The difference in TOA all-sky albedo, multiplied by the CERES TOA incoming solar

flux of 385 W m$^{-2}$, indicates a radiation deficit of $\sim$8 W m$^{-2}$ at a cloud fraction of 0.4, and $\sim$5 W m$^{-2}$ at a cloud fraction of 0.65, for a smokier BL compared to a less smoky BL. This range of direct aerosol radiative effects is a slight underestimation, as we are comparing more smoky conditions with less smoky conditions, instead of the pristine conditions. Even with such crude estimations, we are able to conclude that, for a more smoky boundary layer in the remote SE Atlantic, the CERES TOA





all-sky albedo is reduced in August when smoke is present, meaning radiative impacts from aerosol-cloud-interactions exceed those from the cooling from direct aerosol radiative effect alone, or in other words, aerosol-cloud-interactions dominate the aerosol-only effects within the regional radiation budget.

## 7   Free-tropospheric conditions and meteorological influences

Most fire emissions occur on the African plateau, ~1000 meters above sea level, with isentropic flow initially placing the biomass burning aerosols outflow above the low cloud deck. Thus, much of the smoke within the boundary layer enters from the free-troposphere, through entrainment. A comprehensive visual inspection of the one-minute micropulse lidar extinction and depolarization profiles indicates that elevated smoke plume are frequent above Ascension in August. The lidar discerned elevated smoke layers above Ascension on every day that it was able to sample the free-troposphere (20/23 days in August 2016/2017), multiply-layered at times up to five km. The presence of free-tropospheric smoke is consistent with AERONET-derived aerosol optical depth measurements (Zuidema et al., 2018). Details of the vertical extinction profiles are uncertain enough that the detection of aerosol within the critical 100 m above cloud top is unreliable, but nevertheless the constant presence of free-tropospheric smoke indicates that conditions favorable for entrainment are likely to result in more smoke within the boundary layer, if perhaps displaced from the island.

The inversion strength is examined as a function of the smoke loading in the boundary layer in Fig. 18, to intuit entrainment behavior under different conditions. The relationship of the inversion strength to the smoke loading changes with the smoke loading itself: when the boundary layer is relatively clean (rBC mass concentrations of less than 100 ng m$^{-3}$), more smoke near the surface is associated with a stronger inversion (correlation coefficient of 0.298; Fig. 18, blue dots and fitting curve). This suggests a free-tropospheric semi-direct effect could dominate the cloud response under these conditions, when the boundary layer semi-direct effect is weaker. In contrast, when the boundary layer smoke loading is already high (greater than 500 ng m$^{-3}$), the inversion strength decreases significantly, from approximately 7 K in the mean to almost 4 K (correlation coefficient of -0.5; Fig. 18, red dots and fitting curve). This exceeds the reduction expected from a warming boundary layer alone (of 0.5 K, based on Fig. 13a). An example of this behavior is shown in Fig. 19, in which a lidar-derived aerosol extinction profile is well colocated with the potential temperature profile. This example supports the idea that cooler, well-mixed $\theta$ profiles within individual soundings (Fig. 11) can indicate the presence of overlying smoke. This is evidence that meteorological influences dominate the inversion strength, through the free tropospheric temperature, rather than the shortwave absorption by smoke.

The wind structure associated with the cooler layers will also facilitate entrainment. In the previously-examined mid-August 2016 event, strong easterlies in the lower free troposphere transported the continental smoke efficiently near the top of the marine boundary layer Zuidema et al. (2018). Here the vertical wind structure of the lower atmosphere (up to 3 km) is assessed more comprehensively, as wind variations will affect not only entrainment, but also the overall aerosol and moisture environment. The two years are shown separately, as they are slightly dissimilar. Overall, the winds are the strongest in the surface mixed layer from 200 m to 700 m, with a mean speed of approximately nine m s$^{-1}$. The winds decrease above the ML$_{SB}$ up to the trade-wind inversion base (Fig. 20 c and g). The sub-cloud layer winds are east-south-easterly, with winds veering to the



east above (Fig. 20 d and h). Winds are stronger in the boundary layer and lower free troposphere when the smoke loading is high, consistent with a more "spun-up" circulation (Adebiyi et al., 2015). At 2 km, just above the inversion top, easterlies and north-easterly winds are also more frequent when the boundary layer is smoky, indicating a more efficient transport of smoke (when present) from the continent. More frequent easterlies means more biomass-burning aerosols are brought to the free tro-

posphere above Ascension. Stronger winds as well as an easterly momentum flux at the inversion top encourage entrainment of more smoke into the boundary layer.

## 8 Positive feedback mechanisms

A positive feedback can contribute to the longevity of the week-long smoke events observed at Ascension during August (Fig. 1 and 5), illustrated within a schematic diagram (Fig. 21). On smokier days, in the morning, boundary layers are deeper and

cloud tops are higher. Stronger winds at the inversion top (Fig. 20 c and g) also favor mixing across the inversion. Meanwhile, winds in the free troposphere shift to easterly/northeasterly (Fig. 20 d and h), favoring the transport of smoke off of the African continent (Adebiyi and Zuidema, 2016). The inversion strength decreases (Fig. 18). Boundary layer warming from shortwave absorption is not sufficient to produce a ∼3 K decrease in the inversion strength (Fig. 18), but rather, the well-mixed layers with cooler temperatures directly above the inversion tops explain most of the weakened inversion strength, reflecting a

convolving large-scale meteorological influence. A weaker inversion favors aerosol entrainment if smoke is present overhead and the development of a deeper boundary layer, as more air from the free troposphere is entrained into the boundary layer. In addition, the pronounced moisture stratification within the boundary layer (Fig. 21 $q_v$ profiles, based on Fig. 12 and Fig. 13) during the night indicates clear decoupling, with more moisture trapped near the surface at night. At sunrise, the trapped moisture is ventilated upward, favoring the development of thicker clouds with higher cloud tops, and deepening the BL, further

encouraging smoke entrainment from the free troposphere (Fig. 21 second box). Precipitation suppression by the aerosol (Fig. 5 and Fig. 8) also helps conserve the aerosol loading in the boundary layer, by reducing aerosol removal through wet deposition.

## 9 Synopsis and discussion of the diurnal cycle

We show an overall reduction in cloudiness and in the all-sky top-of-atmosphere albedo when shortwave-absorbing aerosols are present in the marine boundary layer, using a combination of ground-based observations from Ascension Island and areal-

averaged satellite retrievals. These observations indicate that the presence of smoke introduces a profound departure to the "typical" diurnal cycle of marine low clouds, in which nighttime radiative cooling strengthens cumulus-under-stratocumulus, producing a nighttime maximum in LWP and precipitation, and a smaller afternoon cloud and precipitation maximum reflects surface-driven cumulus. This diurnal cycle occurs in the vicinity of Ascension Island when the near-surface smoke loadings are low (e.g, Fig. 12). When more smoke is present, an altered diurnal cycle is summarized in the schematic of Fig. 21.

Daytime thinning and dissipation of the stratiform layer near the inversion base by shortwave absorption is more pronounced, consistent with Johnson (2005), while the daytime warming of the sub-cloud layer reduces the sub-cloud relative humidity,





discouraging cumulus cloud growth. The upper half of the boundary layer becomes more stabilized, and an already-prevailing radiatively-induced afternoon minimum in cloud cover becomes more pronounced and lengthened (Ackerman et al., 2000; Brill and Albrecht, 1982). This is particularly pronounced in August of 2016, attributed to the unprecedented high-smoke event observed during the 2016 mid-august period, when the rBC mass concentration values are almost double the peak value of

August 2017. Although a deeper sub-cloud layer in the afternoon should reduce the $\mathrm{ML}_{sb}$ $q_v$ through diffusion, this doesn't seem to be case, but is consistent with stronger surface winds.

After sunset, a warmer, shoaling sub-cloud layer accumulates the moisture from turbulent fluxes off of the ocean surface beneath a drier upper layer, consistent with reduced turbulence (Pistone et al., 2016; Wilcox et al., 2016). The nighttime cloud cover recovers as an upper-level stratiform layer, but the liquid water paths remain low, indicating the cloud layer remains thin.

This contrasts starkly to the nighttime maximum in cloud cover, LWP and precipitation that is more typical of stratocumulus and some shallow oceanic cumulus regimes, in which the nighttime longwave radiative cooling encourages recoupling of the boundary layer, aided at times by precipitation (Brill and Albrecht, 1982; Miller and Albrecht, 1995; Miller et al., 1998; Nuijens et al., 2009). Instead, the longwave cooling from the thin reforming stratiform layers below the inversion base does not appear to be able to generate buoyant eddies capable of reaching the lower mixed layer. The nighttime cloud cover remains less than

that prevailing under less smoky conditions. The sub-cloud layer, though warmer overall relative to "less-smoky" conditions, cools down throughout the night, equilibrating with the temperature of the upper cloud layer (Fig. 13).

A new observation is boundary layer deepening with higher cloud tops in the morning, with cloud top heights rising and liquid water paths increasing. These observations, combined with evidence of a more well-mixed boundary layer, suggest that radiatively-induced warming initially encourages upward vertical motion. Such behavior has not been documented before, to

our knowledge, and deserves more discussion. To first order, in this equatorial region lacking a Coriolis force, the vertical ascent ($\omega$) can be related to the diabatic radiative warming ($Q$) through $\omega = \frac{Q}{\sigma}$, where $\sigma$ is the static stability. The potential temperature lapse rate within the 200–600 m layer is 0.835 K km$^{-1}$ under smokier conditions (compared to 1.37 K km$^{-1}$ within the less smoky composite). A difference in the radiative heating of the sub-cloud layer from a higher smoke loading of one K day$^{-1}$ can support a vertical ascent of 50 m hour$^{-1}$, so that four hours are required for a clear-sky boundary layer to

deepen by 200 m. The upward motion also ventilates moisture, and liquid water paths increase. Buoyancy generated from the latent heat release from condensation deepens the boundary layer further. The suppression of precipitation from the increased cloud condensation nuclei concentrations provided by the smoke, with more of the smaller drops from the overshooting cloud tops evaporating above the inversion base (Stevens et al., 2005; Seifert and Beheng, 2006), further aids the boundary layer deepening. The enhanced vertical ascent ends by noon, when shortwave absorption has restabilized the upper boundary layer,

and more of the radiative warming remains within $\mathrm{ML}_{sb}$ (Fig. 12b). The near-surface warming never fully dissipates, with the diurnal-mean $\mathrm{ML}_{SB}$ warmer by approximately 0.5 K compared to its "less-smoky" counterpart.

The meteorology that encourages the aerosol transport over the remote southeast Atlantic also supports the entrainment of the smoke into the boundary layer. When the smoke loading is high in the boundary layer, the free-tropospheric winds are typically stronger and more easterly. The advected cooler, better-mixed free-tropospheric potential temperatures weaken the

cloud top inversions at high smoke loadings (rBC mass concentrations > 500 ng m$^{-3}$), indicating a convolving meteorological





influence. The stronger winds and weaker inversions enhance the entrainment of free-tropospheric smoke into the already smoky boundary layer, providing a positive feedback that may help explain the persistence of smoke events within the boundary layer, capable of exceeding a week (Fig. 1 and 5).

The direct aerosol radiative impact is estimated as a cooling of $\sim$8 and $\sim$ 5 W m$^{-2}$ at cloud fractions of 0.4 and 0.65. Despite
this, the net CERES TOA all-sky albedo reduces when the boundary layer smoke loading increases, because of the reduction in the low cloud. Adebiyi et al. (2015) showed that the September-October mean top-of-atmosphere all-sky albedo from CERES increases for more polluted conditions in the SE Atlantic region, despite the frequent presence of the shortwave absorbing aerosols. They argue that this increase in TOA all-sky albedo is caused by an overall increase in cloudiness beneath the smoke layer, consistent with the cloud thickening documented in Wilcox (2010). Further work will be required to fully puzzle out the
evolution of the dominating aerosol-cloud interactions over the course of the biomass-burning season. We also note that this study does not explicitly decompose the likely aerosol-cloud interactions. More in-depth process modeling studies, facilitated with regional cloud-resolving model simulations, will be needed to attribute the changes in low-cloud cover to different aspects of the aerosol-cloud interactions, and to isolate meteorological influences (e.g. fire-on verses fire-off simulations) from the aerosol-cloud interactions. These modeling studies will also benefit from the vertical structure measurements made by
the complementary NASA ORACLES (ObseRvations of Aerosols above CLouds and their intEractionS) and UK CLARIFY (Cloud Aerosol Radiation Interactions and Forcing - Year 2017) aircraft campaigns.

*Data availability.* All the data are publicly-available through the Atmospheric Radiation Measurement Data Archive. The MPL extinction profiles are available upon request.

**Appendix A**

Among the 357 successfully launched radiosonde profiles during August 2016 and August 2017, two were aborted below 3000 meters, and thus, they were excluded from the subsequent analyses. Further quality controls revealed 2 bad wind profiles and 30 profiles that had an issue in relative humidity measurement (anomalously low RH, less than 10 %) in the first 500 m of the boundary layer, however, other variables associated with those profiles turned out to be valid, which were kept in the analyses. Out of those 355 profiles included in the analyses, 114 profiles were identified as daytime soundings and 126 were identified as
nighttime soundings. 0530 UTC (0430 LST) and 1730 UTC (1630 LST) launches were treated neither daytime nor nighttime to avoid potential confusion on whether the boundary layer was sunlit or not. The conditional composite method discussed above yields a total of 92 smoky radiosonde profiles (33 daytime and 38 nighttime) and 63 less smoky profiles (24 daytime and 23 nighttime) in 2016, and 29 smoky radiosonde profiles (7 daytime and 8 nighttime) and 27 less smoky profiles (7 daytime and 9 nighttime) in 2017, out of the 355 profiles. The radiosonde launching time was recorded in UTC, but shown here in LST,
which is approximately one hour earlier, to be consistent with the time stamps of the other measurements. The calculation of



the lifting condensation levels follows equations (4.6.23) to (4.6.25) in Emanuel (1994) using an air parcel originating from 1000 hPa.

*Author contributions.* JZ and PZ conceived the study. PZ acquired the funding. JZ analyzed the data with help from PZ. JZ wrote the paper with reviews and edits from PZ.

5   *Competing interests.* The authors have no competing interests.

*Acknowledgements.* This work was supported by DOE ASR grant DE–SC0018272. The LASIC field campaign was funded by the U.S. Department of Energy's Office of Science, Office of Biological and Environmental Research as part of the Atmospheric Science Research Program. We thank the whole LASIC science team for their efforts on deploying and maintaining the instruments, and the instrument mentors for processing and calibrating the campaign datasets. The MODIS datasets are available through the NASA Level-3 Atmosphere Archive
10   & Distribution System (LAADS) Distributed Active Archive Center (DAAC). The CERES and SEVIRI retrievals are available from NASA Langley Research Center. We thank Douglas Spangenberg for their development.



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



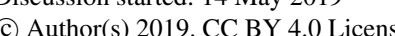

Figure 1. Time series of single-particle soot photometer (SP2)-derived refractory black carbon mass concentrations from 1 June to 31 October for a) 2016 and b) 2017.





**Figure 2.** Radiosonde-derived monthly-mean potential temperature, water vapor mixing ratio and relative humidity profiles from the surface to three kilometers altitude in colored curves, for 2016 (a, b, and c) and 2017 (d, e, and f). The median,10%, 25%, 75% and 90% percentiles of August-mean cloud top inversion base and lifting condensation level are indicated as black and red box-whiskers, respectively, on all panels, with filled circles indicating the mean.



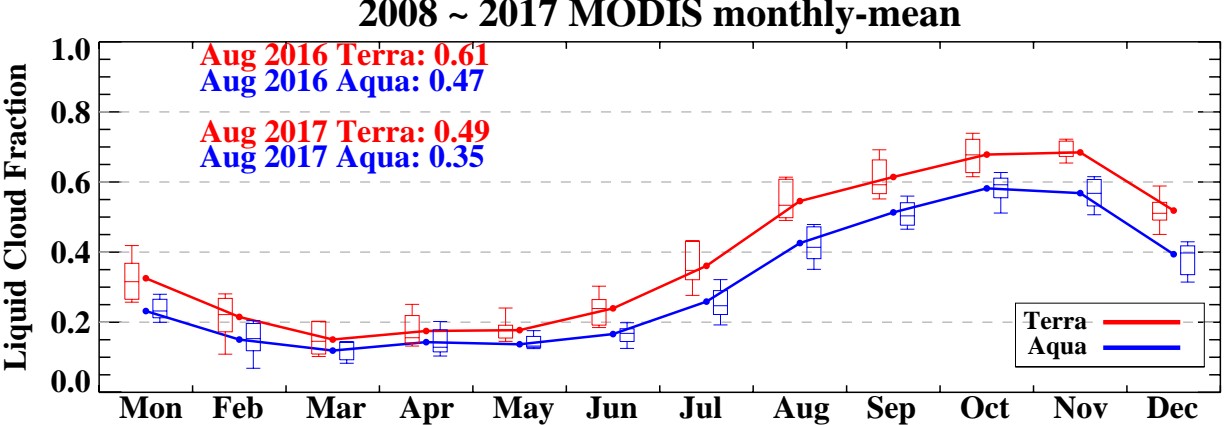

**Figure 3.** Annual cycle of MODIS-derived monthly-mean liquid cloud fractions averaged over a 4°by 4°box slightly to the east of Ascension (6°-10°S, 11°W-15°W) from 10 years of data (2008 – 2017). The 10th, 25th, 75th and 90th percentiles are indicated by the box-whiskers, and mean values are shown using filled circles connected with lines. Red and blue colors represent the Terra and Aqua satellites respectively.





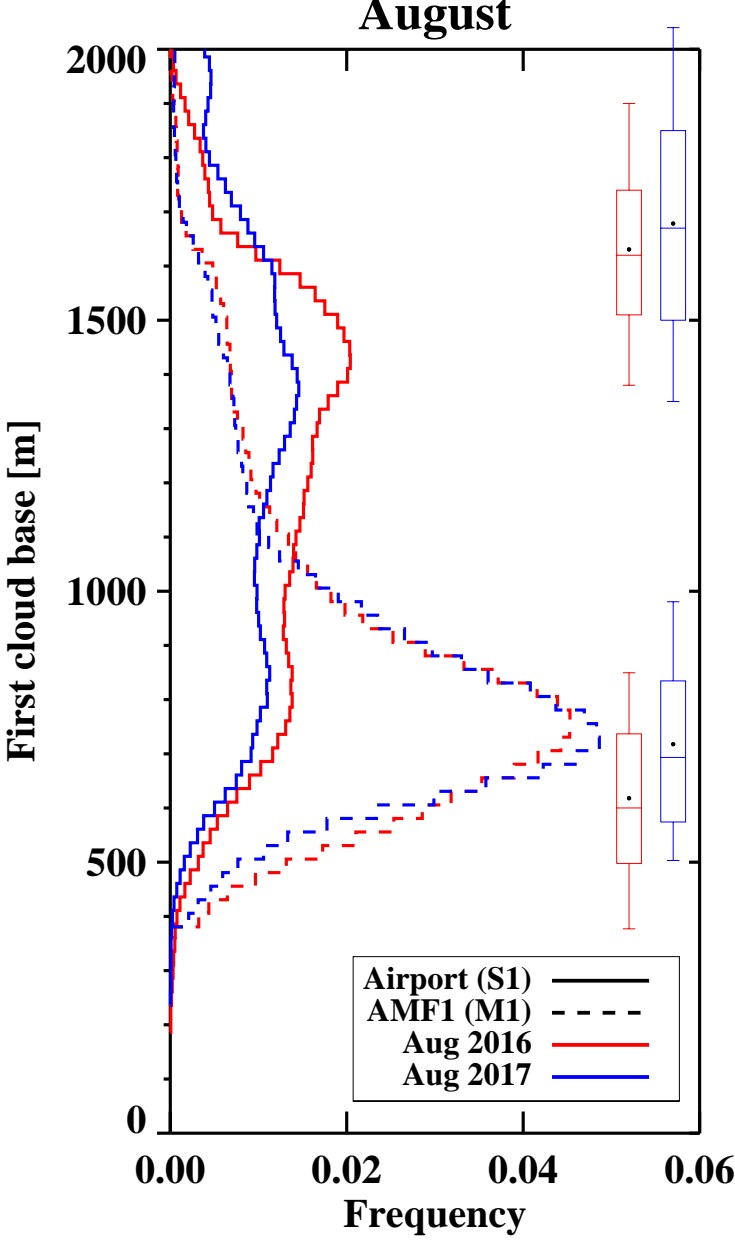

**Figure 4.** Normalized frequency distribution of the lowest ceilometer-derived cloud base for August 2016 (red) and 2017 (blue) at the airport (solid) and AMF1 site (dashed). The median,10%, 25%, 75% and 90% percentiles of the August radiosonde-derived lifting condensation level and trade-wind inversion base are indicated as box-whiskers for both years, with filled circles indicating the mean.

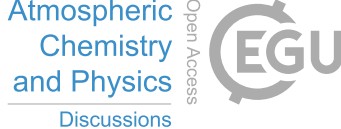

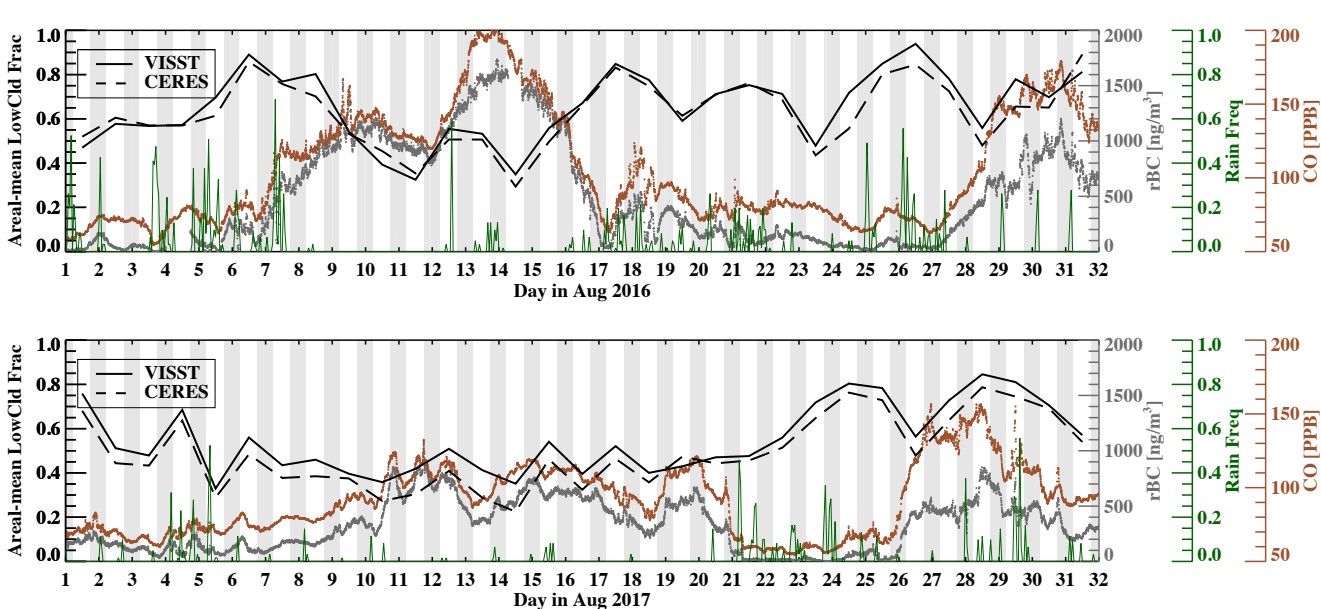

**Figure 5.** Time series of refractory black carbon mass concentration (gray filled circles), carbon monoxide concentrations (brown filled circles), disdrometer-derived hourly rain frequencies (dark green curves) at the AMF1 site, and daily low-cloud fractions averaged over 6°-10°S, 11°W-15°W for August 2016 (upper) and 2017 (bottom). The daily-mean Meteosat10 VISST cloud fractions are indicated in solid lines, and the CERES Terra and Aqua low-cloud fractions are in dashed lines.

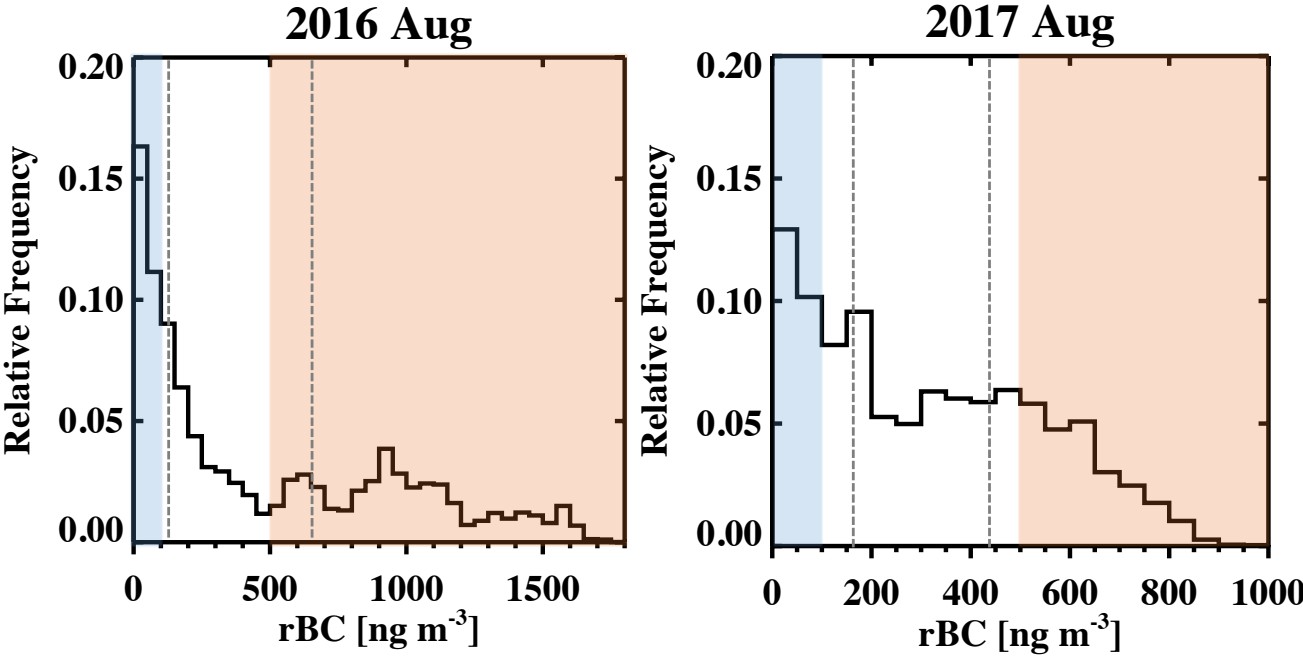

**Figure 6.** Histogram of one-minute refractory black carbon mass concentration at Ascension Island for August 2016 and 2017. Red and blue shadings indicate "more smoky" and "less smoky" conditions, respectively, and tercile values are indicated with gray dashed lines.




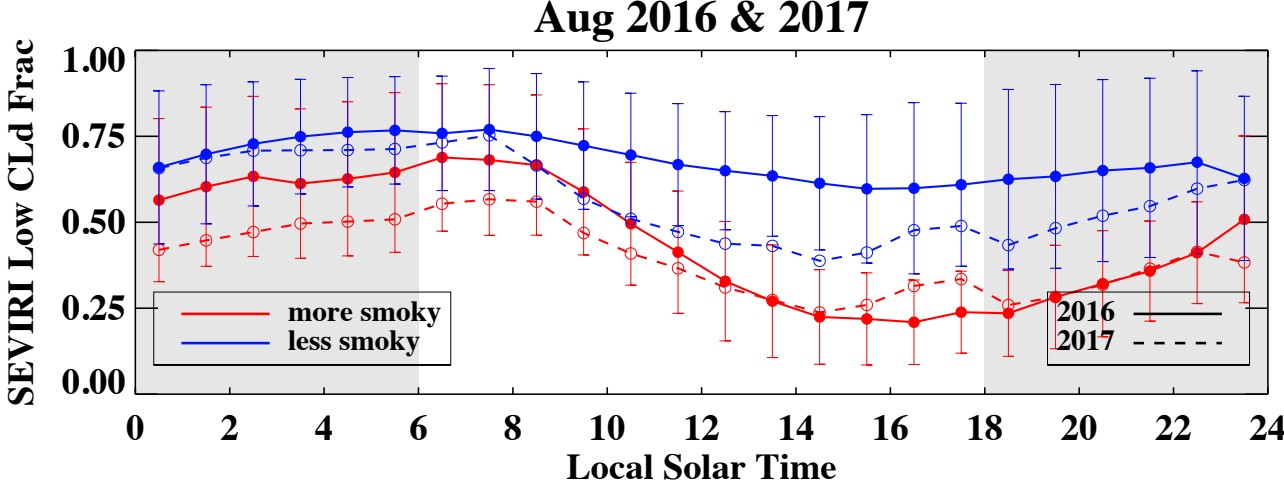

**Figure 7.** SEVIRI-derived diurnal hourly low cloud fraction, composited by "more smoky" (red) versus "less smoky" (blue), for August 2016 (solid line) and 2017 (dashed line). The means are connected by solid lines with error bars indicating one standard deviation.



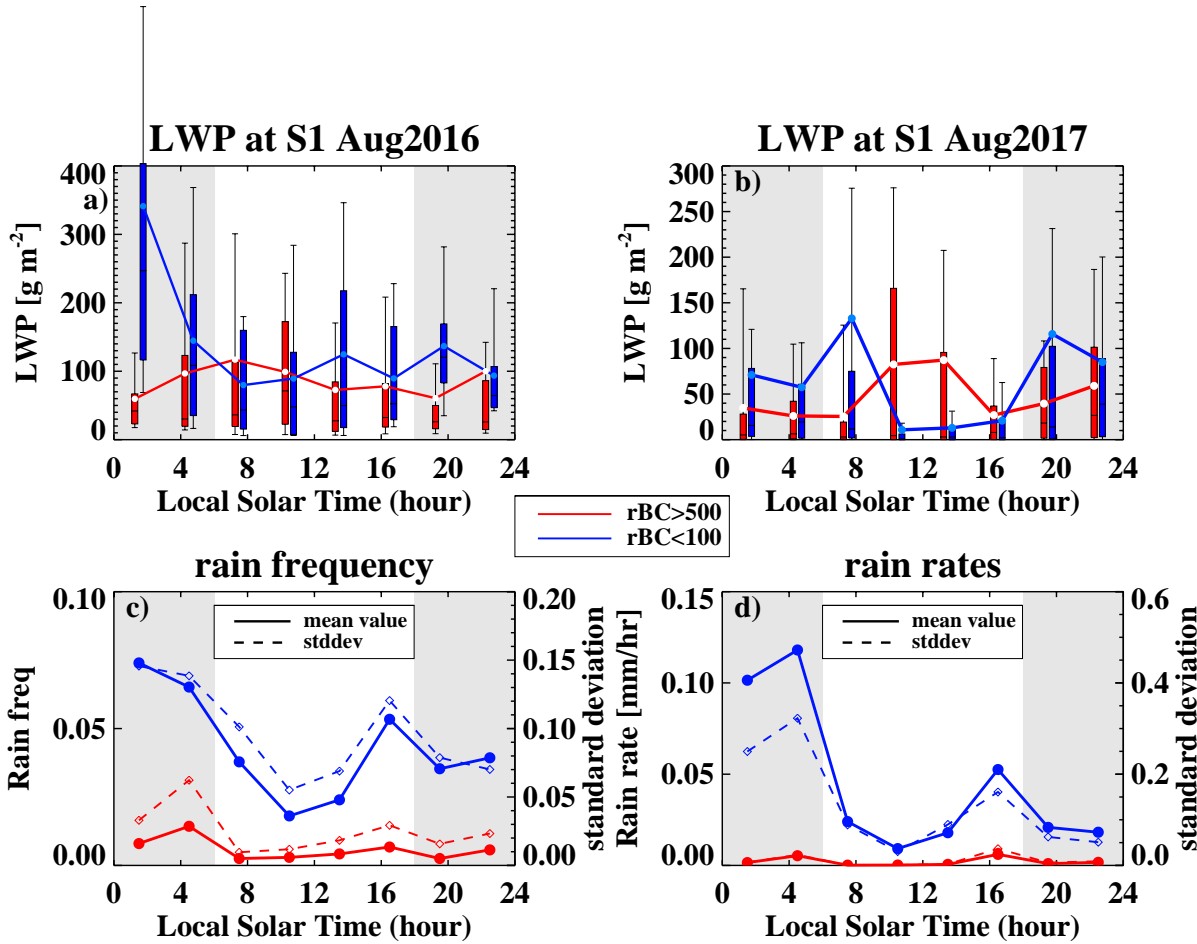

**Figure 8.** Diurnal cycle in the liquid water paths at the airport for a) August 2016 and b) August 2017, composited by "more smoky" (red) versus "less smoky" (blue), with box-whiskers indicating the 10, 25, 50, 75, and 90th percentiles, and filled circles the mean values. c) Diurnal cycle in disdrometer-derived rain frequencies and d) rain rates at the AMF1 site, similarly composited, shown using averages and standard deviations (solid and dashed lines). Only one-minute samples with rain rates exceeding 0 mm/hr are included. The diurnal cycle is depicted throughout using three-hour averages (0-3, 3-6, 6-9, 9-12, 12-15, 15-18, 18-21, 21-24 LST) plotted at their midpoint in time. Gray shadings indicate night time.



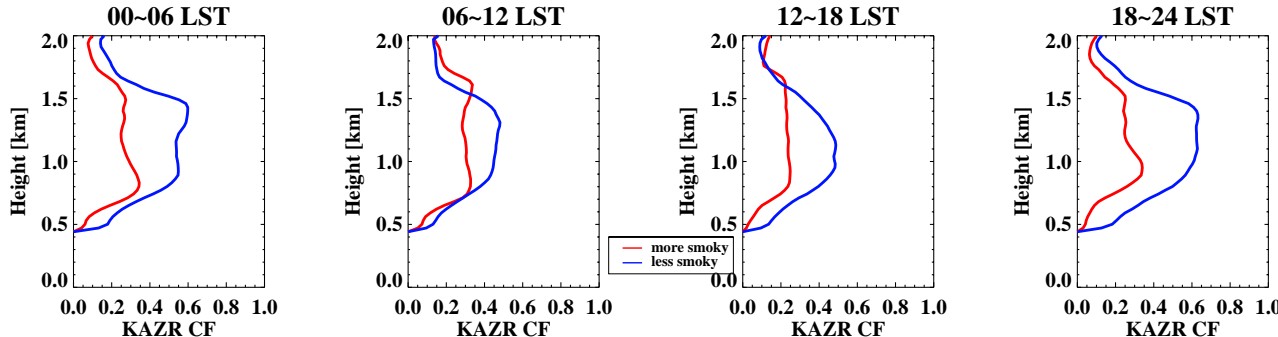

**Figure 9.** Diurnal cycle in cloud fraction as a function of altitude derived from the Ka-band zenith pointing cloud radar (KAZR), depicted every 6 hours, composited by "more smoky" (red) versus "less smoky" (blue) conditions.



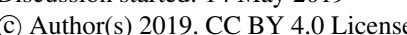

**Figure 10.** Hourly-mean SEVIRI retrievals of effective cloud top heights as a function of low-cloud fractions over 6°-10°S, 11°W-15°W. Cloud top heights corresponding to "more" and "less" smoky conditions are indicated in red and blue, respectively, and include linear fits as a function of cloud fraction. 08 LST samples are highlighted in green crosses regardless of smoke loading.

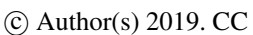



**Figure 11.** a) and d) potential temperature, b) and e) relative humidity, and c) and f) water vapor mixing ratio radiosonde-derived vertical profiles for August 2016 (upper) and 2017 (bottom), shown individually and as composite means, colored by "more" (red) and "less" (blue) smoke loadings. Composite means are indicated by thicker curves, daytime and nighttime by solid and dashed curves. The smoke loadings are based on hourly-mean rBC values centered on the radiosonde launch time. 92 and 63 profiles contributed to the "more" and "less" smoke loading distribution, respectively, in 2016, and 29 and 27 profiles contributed similarly in 2017.





**Figure 12.** Three-hourly August-mean potential temperature profiles from a) 100 m to three km, and b) 100 m to one km, c) water vapor mixing ratio profiles from 100 m to one km, and d) relative humidity profiles from 100 m to one km. All are depicted as averages over "More" (red) and "Less" (blue) smoke loadings. 08 LST profiles are highlighted with thicker lines.



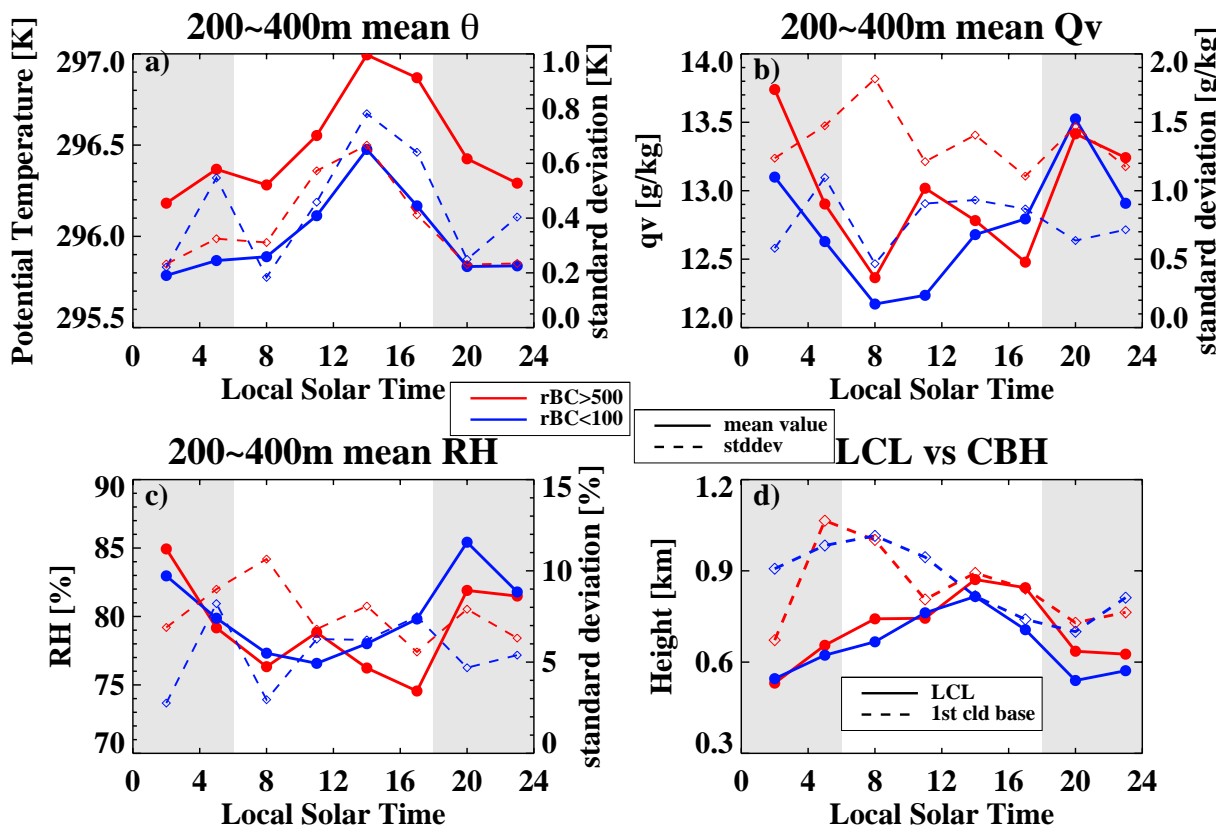

**Figure 13.** Three-hourly radiosonde-derived 200–400m mean a) potential temperature ($\theta$), b) water vapor mixing ratio (Qv), and c) relative humidity (RH), d) radiosonde-derived lifting condensation levels (LCLs) and the 10th percentile of the lowest ceilometer-derived cloud base. All are composited by "more" (red) and "less" (blue) smoke loadings. standard deviations indicated with thinner dashed curves and gray shadings indicate night time. Time of day is the same as in Fig. 12.



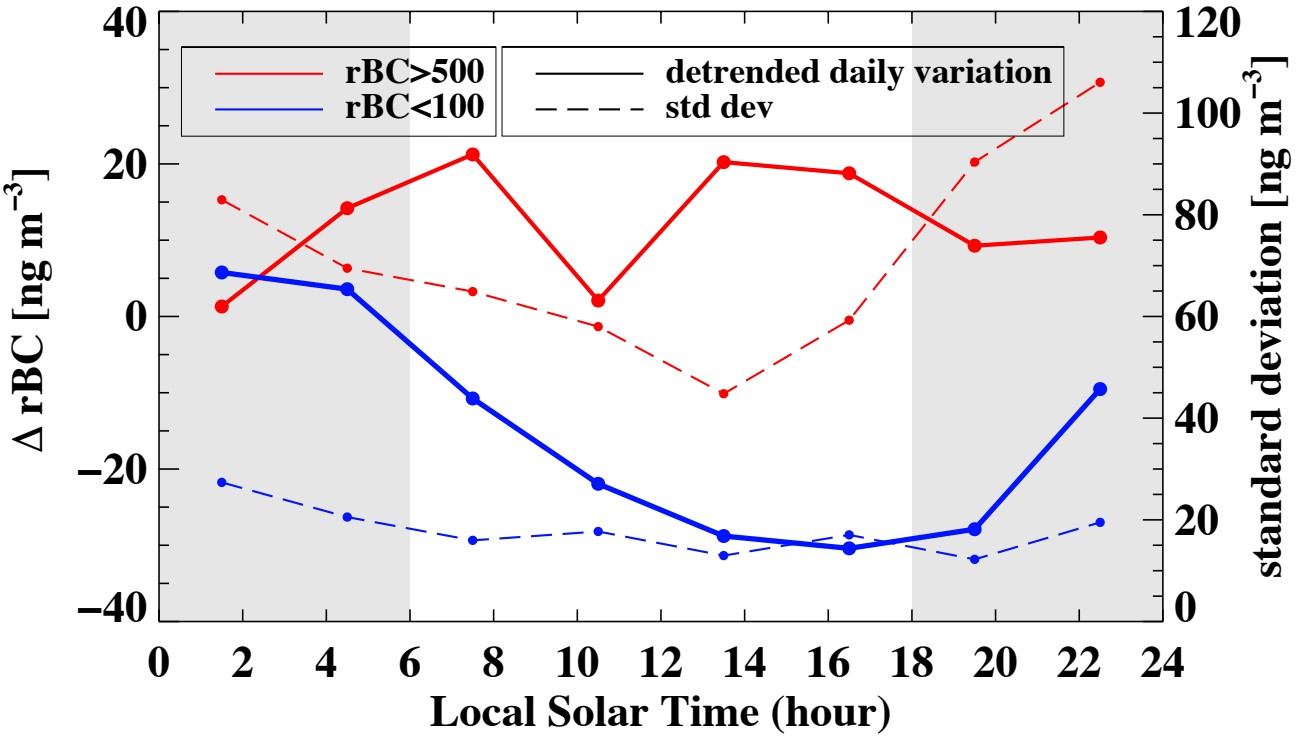

**Figure 14.** Three-hourly averages of the detrended refractory black carbon mass concentrations composited by "more" (red) and "less" (blue) smoke loadings. Days with sharp, short-term changes in rBC loading are excluded. Standard deviations are overlaid in thinner dashed curves. Time plotting conventions are the same as in Fig. 8. Gray shadings indicate night time.







**Figure 15.** First and third rows: daily time series of Ka-band zenith pointing cloud radar reflectivities (colored contour), disdrometer-derived one-minute rain rates (magenta) and one-hour rain frequencies (dark green), for a) August 12, 2017 and b) August 20, 2017. Ceilometer detected lowest cloud bases are overlaid in black dots. Terra and Aqua overpass time are indicated in red dashed lines. Second and fourth rows: corresponding 2°by 2°MODIS Terra and Aqua visible imagery centered on Ascension.







**Figure 16.** left) potential temperature, middle) water vapor mixing ratio, and right) relative humidity radiosonde-derived profiles for a) August 12, 2017 and b) August 20, 2017. Daily-mean black carbon mass concentration are indicated on the left panel, and individual radiosonde launch times within the legend.





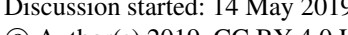

**Figure 17.** Daily and areal-mean all-sky top-of-atmosphere albedo as a function of corresponding low-cloud fraction, both from CERES. "More" and "less" smoke loadings indicated in red and blue, respectively, with linear fits and correlation coefficients also indicated. Inset: albedo histograms.





**Figure 18.** Radiosonde-derived inversion strength as a function of the refractory black carbon mass concentrations. "more" and "less" smoke loadings indicated in red and blue, respectively. Linear fits and correlation coefficients included for the two composites. Note the logarithmic x-axis.





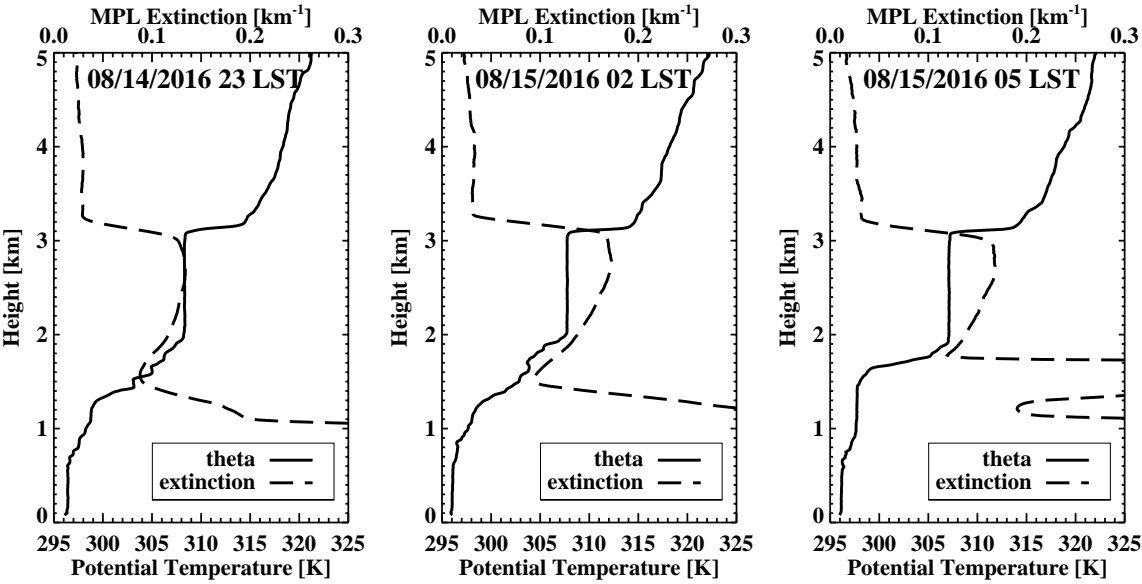

**Figure 19.** Potential temperature (solid, from radiosonde) and volume extinction coefficient (dashed, derived from micropulse lidar following Delgadillo et al. (2018) from the one hour averaged around radiosonde launch time) profiles for a) August 14, 2016, 23 LST, b) August 15, 2016, 2 LST, and c) August 15, 2016, 5 LST.





**Figure 20.** Radiosonde-derived a) and e) zonal wind profiles, b) and f) meridional wind profiles, and c) and g) mean wind speed profiles, for August 2016 (upper panel) and August 2017 (lower panel). Solid thick curves indicate composite-mean, and horizontal bars indicate 10% and 90% percentiles. Inversion bases and tops indicated using mean values (filled circles), and 10%, 25%, 75% and 90% percentiles (black box-whiskers) in c) and g). d) and f): wind roses indicating wind directions at 500m, one and two km. "More" and "less" smoke loadings indicated in red and blue respectively.

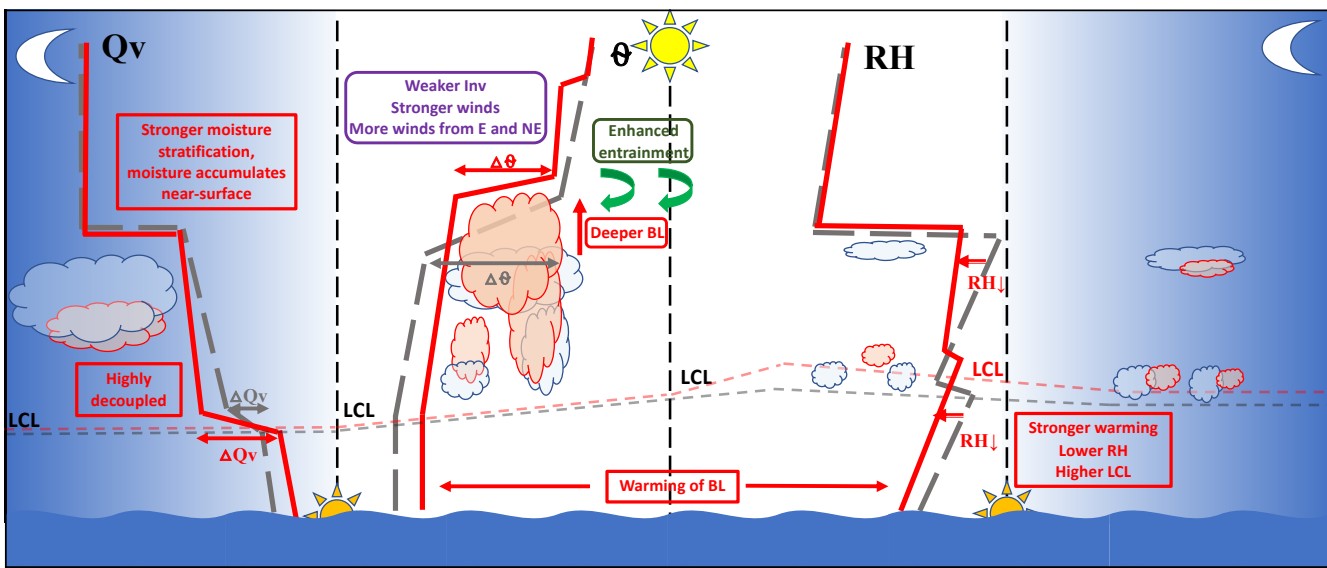

**Figure 21.** Schematic of the diurnal cycle under "less" (blue) and "more" (red) smoky conditions.



**Table 1.** Full list of datasets used in this paper with source/contact and remarks. The airport site is S1 and the AMF1 site is M1.

| Name | Full Name | Uncertainties | contact | remarks |
|---|---|---|---|---|
| SONDE (S1) | Balloon-Borne Sounding System | 0.5 °C and 5% RH | D. J. Holdridge, J. A. Prell, M. T. Ritsche, and R. Coulter | Vaisala, Inc. RS-92 |
| CEIL (S1 and M1) | Vaisala Laser Ceilometer | cloud height measurement sensitivity: 10 m | V. Morris | Vaisala, Inc. CL31 |
| SP2 (M1) | Single Particle Soot Photometer | 3 ng m$^{-3}$ | A. J. Sedlacek | Droplet Measurement Technologies, Inc. |
| CCN (M1) | Cloud Condensation Nuclei Particle Counter | activated particle sizing $\pm 0.25$ $\mu$m | G. Senum and J. Uin | Dual-chamber, values shown at 0.4% SS from Column A |
| MPL (M1) | Micropulse Lidar | see Delgadillo et al. (2018) | P. Muradyan | Sigma Space Corporation |
| DIS (M1) | Impact Disdrometer | range of diameter: 0.3 mm to 5 mm | M. J. Bartholomew | Distromet LTD Basel, Switzerland |
| SEVIRI | Spinning Enhanced Visible and Infrared Imager | N/A | NASA Larc | Visible Infrared Solar-Infrared Split Window Technique |
| CERES | Clouds and the Earth's Radiant Energy System | N/A | NASA Larc | Synoptic TOA and surface fluxes and clouds (SYN) products onboard Aqua and Terra |
| MODIS | Moderate Resolution Imaging Spectroradiometer | N/A | NASA | level-3 one-degree Onboard Terra and Aqua |
| CO (M1) | Carbon Monoxide Analyzer | $\pm 2$ ppbv | S. Springston | Los Gatos Research |
| KAZR (M1) | Ka-band zenith pointing cloud radar | reflectivity copol $\sim 3$ dBZ | N. Bharadwaj | -35 dBZ cutoff for clouds |
| MWR and MWRP (S1) | Microwave Radiometer (Profiler) | Tb $\sim 1$ K LWP $\sim 0.015$ mm | M. P. Cadeddu | MWRRET2 physical retrievals (23.8 GHz and 31.4 GHz) |