# Peer review of "The diurnal cycle of the smoky marine boundary layer observed during August in the remote southeast Atlantic"

_Atmospheric Chemistry and Physics, 2019_

## Referee Comment (RC1) · Anonymous Referee #1 · 24 Jun 2019

Review of "Low cloud reduction within the smoky marine boundary layer and the diurnal cycle" by J. Zhang and P. Zuidema

This manuscript presents a characterization of smoky and non-smoky boundary layer (BL) structures and their diurnal cycle over the Ascension Island. Measurements from the years 2016 and 2017 show that high levels of smoke frequently are observed in the BL in the month of August. Simultaneously, the observed BL is warmer and deeper and the cloud fraction is lower. The authors argue that there is a causal relationship between the increased amounts of smoke and the lower cloud cover fraction and suggest a potential mechanism.

The presented observations of the BL over the Ascension Island are interesting and highly relevant for the scientific community. However, I find several issues with the manuscript in its current form. Firstly, and most importantly, the suggested causal relation between the enhanced smoke levels and cloud fraction is speculative and not necessarily supported by the observations. The causal relation is an hypothesis and should be presented as such. In fact, I find that relationship could very well work in the other direction: that it is the BL structure, cloudiness and exchange with the free troposphere that results in enhanced levels of smoke in the BL. The authors should at least consider this potential explanation. Secondly, the overall message, structure and clearness of the manuscript could be substantially improved.

Main comments:
- Throughout the manuscript, it is assumed that elevated BC concentrations *cause* certain changes in cloud cover, precipitation, temperature, BL structure etc. But in most (all?) cases the relation could very well be the opposite. For example, in Figure 7, it seems to me that the blue curves (in particular for 2016) show a diurnal variation in cloud fraction typical for a Sc deck while the red curves are more typical for a situation with Sc to Cu transition (cf. e.g. the review on stratocumulus by Wood, 2012). Could it not be that the overall large-scale meteorological conditions during certain periods are more favorable for Sc to Cu transition and afternoon Cu formation, i.e. that the subsidence is weaker? This would also favor a weaker inversion and more mixing of free tropospheric (polluted) air into the BL. Other examples:
  - Page 7, lines 5-6: "A boundary-layer semi-direct effect is clear: when more smoke is present in the boundary layer, the diurnal-mean low-cloud cover decreases." To me, the relationship could very well work the other way.
  - Page 7, lines 21-22: "The presence of more smoke reduces how frequently rain reaches the surface, although a slight late-night maximum and secondary late-afternoon maximum is still present." Rain could also decrease the smoke concentrations.
  - Page 8, lines 21-24: "Diurnally-resolved mean thermodynamic profiles indicate that the deepening of the smokier boundary layers occurs during the morning, followed by more significant shoaling during the night (Fig. 12a). The potential temperature of the smokier composite is the most warm near the surface in the mid-afternoon (14 LST), with the warmth moving upward in the late afternoon (17 LST)." What is the cause and the effect here?
  - Page 8-9, lines 30 and 1-2: "The warming is greater near the surface than higher in the boundary layer, which suggests that the absorbing aerosol may be preferentially located near the surface." Alternatively, the surface is just warmer in these cases.

- Related to the comment above, I miss a discussion on the general BL structure in the region and its diurnal variation. What would be the typical BL variation (and large-scale situation) for a day with thick stratocumulus clouds and a day with more stratocumulus above cumulus? It appears to me that in non-smoky conditions, the surface layer is stably stratified during night. In smoky conditions, it stays relatively well mixed (Fig 12). Could this indicate more mixing from above for the smoky conditions?
- Statistical significance, usage of "similar", "different", "clearly less" etc. and general uncertainties: There are in total 13 smoky days and 13 less-smoky days, how robust are the conclusions that you draw from this number of cases? Some examples:
  - Page 7, lines 13-14: "The LWP is clearly less in the afternoon under smokier conditions in 2016, though not so in 2017." Compared to what? And is the difference robust/statistically significant? How many samples do you have in each bin?
  - Page 7, lines 14-15: "A more consistent feature between the two years is the reduction in LWP apparent after midnight in smokier conditions. Also consistent is an atypical increase in LWP that occurs mid-morning, after the cloud cover has begun to decline." Are these differences/variations significant?
  - Figures 8c&d and 9: what is the standard deviation for the different curves? Can you say with confidence that "Clouds are fewer and may be less vertically-developed in smokier boundary layers" and "… cloud tops are higher in the morning (06 – 12 LST) under smokier conditions…"?
  - Figure 13 and the discussion on page 9: what is the standard deviation? "The sub-cloud RH decreases during the day for higher smoke loadings…" but this occurs also for low smoke conditions? And is this decrease really robust? Same with "…with both peaking around 20 LST for light smoke loadings, and several hours later, around 2 am LST, for heavier smoke loadings." Is this really a robust variation?
- Sections 5, 6 and 8: I am not sure these sections add much information. In general, I found them too speculative and did therefore not comment them in detail. In Figure 17, I would say that the differences are not significant.
- Section 7: This section could be extended and written more carefully. For example:
  - "Most fire emissions occur on the African plateau, ~1000 meters above sea level, with isentropic flow initially placing the biomass burning aerosols outflow above the low cloud deck. " How often does this happen? And how would this influence your results/conclusions?
  - "Comprehensive visual inspection of the one-minute micropulse lidar extinction and depolarization profiles indicates that elevated smoke plume are frequent above Ascension in August." How often? This type of statistics would be highly relevant to show.

Minor comments (on the sections I read in more detail):
*Abstract:*
- Line 3: "… recent observations…", specify when.
- Line 13: "…. increase…" compared to what?
- Line 21: "…more easterly…" compared to what?
*1. Introduction*
- I find the structure with unpublished figures in the introduction strange. I would recommend to include these figures (and the associated discussion) in Section 2 instead, as a part of the overview.

- I miss some general question/questions stated at the end of the introduction.
- Line 6: "…stratocumulus thickening can occur…"
- Line 19: "This type of marine…"
- Line 17: What do you mean with "The diurnal time scale is faster than that of meteorology…"?

**2. Data, overview and compositing approach**
- Page 4, line 5: What do you mean with "additional cloud"?
- Page 4, line 22: Define abbreviation (rBC).
- Page 5, lines 3-4: What do you mean with "perhaps through their enclosure"?
- Page 5, line 23: How do you know that the retrieved LWPs include only Sc and shallow Cu?
- Page 6, line 14: "…indicate remarkably similar conditions…" Similar in terms of what? Cloud cover? rBC? CO? or all? And how do you define "similar"?
- Page 6, lines 16-17: "The exception is 21-23 August of 2017, when CO concentrations are close to the background value and rBC mass concentrations are…" What about 2016, 1-6 and 24-27?
- Page 6, line 19: "lasting for just a little shy of a week." Are you talking about 2016 or 2017?
- Page 6, line 21: "…clearly reduces this time…" compared to what?

**3. The cloud diurnal cycle as a function of the smoke loading**
- Page 7, lines 31-32: what do you mean with "…when they correlate more strongly with lower cloud fractions…"?

**4. Explanations for the altered cloud diurnal cycle**
- Page 8, lines 5-6: "The radiosonde thermodynamic profiles (Fig. 11) indicate boundary layer depths that can exceed the cloud top heights documented in Fig. 10." But you don't see all cloud tops in Figure 10?
- Page 8, lines 7-8: "The deepening of the boundary layer is more well-defined…" compared to what?
- Page 8, line 28: I would suggest to change "fluxing" to "flux".
- Page 9, lines 5-6: Days with daily rBC variation less than 120 ng m-3 and 50 ng m-3, respectively, are selected from the "more" and "less" smoky days, and detrended. Please clarify.
- Page 10, line 10: "Under smokier conditions, the near-surface rBC values accumulate after mid-night.". This difference do not seem robust.
- Page 10, lines 12-13: "Overall, during a typical day, smoke settles towards the surface under smokier conditions, and is advected upwards when the surface layer is less smoky". This formulation is unclear to me.

**5. Examples**
- Page 10, lines 20-21: "…indicate cumulus clusters in the morning…" or cumulus under stratocumulus?
- Page 10, line 22: What do you mean with "upper level cloud"?
- Figure 15: what is RF in the figure? And what does the KAZR radar mainly see? I assume it is mostly precip-sized particles, this should be clarified.
- Page 10, line 23: "…with only scattered cumuliform clouds remaining…" How do you see this? And are you talking about both cases?

**References**

Wood, R. 2012. REVIEW: Stratocumulus Clouds. Monthly Weather Review, 140, 2373-2421.

---

## Referee Comment (RC2) · Anonymous Referee #2 · 25 Jun 2019

Zhang and Zuidema present a very detailed and insightful analysis of clouds and boundary layer structure in the Southeast Atlantic ocean for situations with low and large smoke concentrations. A lot of different observations are compiled and analysed in order to describe the differences between the two situations. The study is very useful and very well conducted, and the text and figures are very well prepared.

However, I have two larger concerns, and a number of minor concerns, that the authors should address before publication.

1. Causality. At several instances the text suggests a causal link from smoke to clouds and precipitation that is not obvious. The general solution is to describe what is seen and not to assert causality where it is not evident. The most important aspect here is the finding that there is less precipitation when there is more smoke. I am convinced
that here the causality is that when it rains, smoke is washed out. The authors propose the opposite causality, i.e. that rain is suppressed due to the smoke. Unless one applies models, it is impossible to prove either causal pathway. So my suggestion is to just use a different language to describe what the observational result is.

2. Aerosol forcing. In my opinion, the difference in the two lines in Fig. 17 is mostly attributable to differences in LWP, so what the authors infer is not a radiative perturbation due to the aerosol-radiation interactions. There are means to infer the radiative forcing. It seems the authors mostly want to assess the clear-sky scattering effect, which is easy to do using the AOD that should be available e.g. from MODIS (or from Aeronet). The authors could use an albedo scaling with AOD perturbation of 0.07 AOD-1 (Myhre et al. ACP 2013, 10.5194/acp-13-1853-2013). If one was to include the effect of aerosol absorption above clouds, the calculation becomes more difficult (e.g Peters et al. ACP 2011 10.5194/acp-11-1393-2011).

p1 l15 "direct aerosol radiation" is probably better as "increase in reflected solar radiation in clear sky attributable to aerosol" or so?

p2 l3 "decks"

p2 l17 "prior"

p3 l17 The authors should define what they mean by "meteorology" here.

p4 l8 In fact the airport is shown in plain

p5 l3 Here and later, "meter" should be abbreviated as "m". Same later for "seconds" → "s"

p5 l22-23 This sentence I do not understand. One radiometer failed in 2016, and another one was used in 2017? Where is the problem?

p6 l3 The definition "liquid-only water+suspected water" requires slightly more explanation. Is "water" = "liquid cloud"? Or rather open ocean surface?

p6 l10 The difference in the two MODIS retrievals (the one included in the CERES product and the other one used) should be explained, ideally by providing the references for the two retrieval algorithms.

p6 l12 Is the difference between MODIS and SEVIRI due to the fact that SEVIRI resolves the diurnal cycle? i.e. is SEVIRI = MODIS if SEVIRI is sampled at the (four? or more?) MODIS overpass times?

p6 l16 What is the CO concentration background value?

p6 l22 Another obvious explanation is the opposite causality: smoke is low when it is washed out by rain, and larger when there is no rain

p6 l28 What motivates the current choice of thresholds, rather than the actual terciles?

p7 l6 "is clear" - there is no proof for causality here. Why not a more cautious language just stating the co-variation?

p7 l11 "more to a measure of cloud thickness than cloud fraction" - is this in fact all-sky LWP? Because else the sentence is mis-leading: it only is cloud thickness, and not cloud fraction (even if of course ancillary knowledge is that the two are usually correlated).

p7 l21 The causality presumably is the inverse: there is less smoke when it rains

p11 l26 Also LWP differences play a – presumably dominant – role.

p12 l2 So far I would have understood the topic is cloud adjustments to aerosol-radiation interactions ("semi-direct effect"), not so much aerosol-cloud interactions (what was called "microphysical" earlier in this manuscript).

p14 l21 Diabatic warming, not necessarily radiative: precipitation can play a role (and presumably usually does, since mostly radiation acts to cool the layer)

p24 Fig. 3: The median obviously is also shown. (same p25 Fig 4)

[Figure]

p28 Fig. 7: presumably the grey shading indicates sun below horizon?

p35 Fig. 14: slightly more explanation is necessary on how the detrending is done, and on what exactly is the "Delta" value on the y-axis. Why not a deviation from the mean?
* * *

---

## Referee Comment (RC3) · Anonymous Referee #3 · 27 Jun 2019

**1   General comments**

1. The manuscript presents comprehensive observations of the boundary layer from a field campaign in Ascension Island, together with an analysis of the links between smoke and the diurnal cycle of low clouds. The paper is marked by a detailed and careful analysis.

2. The 'Introduction' section is a mixture of state-of-the-art review and motivation. In my view, a better separation of both would be helpful for the reader. After a literature review the open questions to be addressed here should be stated clearly, along with the hypotheses. Also, it did not become entirely clear to me why the diurnal cycle is of particular relevance.

[Figure]

3. The paper is structured into a multitude of sections of markedly differing length. While this may make it easy for a reader to find specific information, it might help to further aggregate sections under unambiguous headings such as 'results' and 'discussion', or at least to make clear in the section headings which belongs to which category. My preference is aggregation.

**2  Specific comments**

1. page 1, line 0 (henceforth 1-0 etc.): In the title, it is not clear what "the diurnal cycle" refers to (clouds, obviously, from the text). Also, it is not clear that the link between smoke and clouds is at the center of this study. Suggestion "Smoke impacts on amount and diurnal cycle of low marine boundary layer clouds in the SE Atlantic" – or something similar.

2. 1-4: awkward sentence. Smale is not only "highly absorbing" in August. Please try rewording.

3. 1-9: What is a "surface-based" mixed layer?

4. 1-11: I don't understand what is meant by "increasing the moisture stratification" – how can a stratification increase?

5. 1-15: This sentence seems confusing and counter-intuitive to me. Absorbing aerosol decreases radiation at TOA, correct? So taking the difference between the least and most smoky terciles will yield a negative figure, i.e. a reduction. I suggest presenting this fact in this way.

6. 4-4: The subsection beginning here seems a mixture of process discussion ("orographic lifting cleary induces...") and a description of the context of the measure-

ments. I suggest moving the former to the state-of-the-art section, and giving the remainder a sub-title (2.1 Context of measurements, or similar)

7. 6-3: "suspected water" – I assume this refers to liquid water?

8. 6-5: apparent LIQUID-water clouds (?)

9. 6-7: How is this evaluated?

10. 6-9: "For context" – what does this mean?

11. 7-3: sunrise and sunset are not indicated in figure 7.

12. 7-3: The peak is barely visible, therefore the reader cannot be sure that this 'peak' actually represents all or most individual days. One solution to this might be to show deviation from the mean cloud fraction of each day (daily cloud fraction anomaly) on the vertical axis instead of the actual cloud fraction.

13. 7-9: See comment on cloud fraction, line 7-3.

14. 8-13: What is a 'surface-based' layer?

15. Fig. 21: I very much like this conceptual summary of your work. However, I ask you to consider the following points: Within each of the four phases, can the blue and red clouds be placed next to each other for visual clarity? 1) In the morning I find it hard to make out the exact borders of the blue clouds. 2) blue is in front of red in the first panel, behind in all others. 3) placing 'mignight', 'noon' and 'midnight' on the horizontal axis might help the reader. 4) the warming of the BL progresses during daytime. The red horizontal arrow in the morning should therefore point to the right. 5) Please explain all abbreviations in the caption.

**3   Technical details**

1. 1-3: "free troposphere" (no hyphen)

2. 1-6: "low-cloud cover" (cover of low clouds, not low cover of clouds). Also in other locations around the manuscript (e.g. 2-8, 6-12)

3. 2-3: deckS

4. 4-4: volcanically produced → volcanic

5. 4-15: remove "local"

6. 4-22: measureMENT

7. 4-22: concentration (no s)

8. 5-2: ...airport, yielding profile measurements of the complete boundary layer in 357 successful...

9. 6-18: August

10. 8-2: larger instead of higher

11. 8-23: warmest instead of 'the most warm'

12. 9-13: 'retrieved' instead of 'inferred'

13. 10-4: 'analyzed' instead of 'interrogated'

14. 12-15: 'infer' instead of 'intuit'

15. 15-1: 'is reduced' instead of 'reduces'

16. Fig. 13: 'means of' instead of 'mean'

17. Fig. 1: introduce 'rBC' in the caption

18. Fig. 10: harmonize size of labels between figures
* * *

---

## Author Comment (AC1) · 29 Aug 2019

**Response to reviews of "Low cloud reduction within the smoky marine boundary layer and the diurnal cycle" by J. Zhang and P. Zuidema**

First of all, we would like to thank the three anonymous reviewers for their constructive comments on our manuscript, which encouraged us to think more deeply about our analysis.

All three reviewers mentioned the writing could be improved in particular Revs. 1 and 3. Rev. 1 and 2 mentioned the text suggests causal links that are not obvious (mostly an overassertion of a more smoke -> less clouds/precip relationship), and Rev. 3 mentioned a lack of an explanation for why the diurnal cycle is relevant. In response, we have restructured the manuscript into fewer sections (from 9 to 6), paid more attention to the language and rewritten the portions that overemphasized a causality we can't clearly show with observations, rewritten the introduction to clarify the separation between literature review and motivation, and rewritten the introduction/summary/discussions to emphasize the importance of the diurnal cycle (for, e.g., the net radiative impact, and, as a model metric).

Reviewers 1 and 2 mention a concern that the smoke->cloud fraction relationship is overly speculative and not necessarily supported by the observations. We certainly acknowledge that the opposite relationship, of aerosol removal through wet scavenging by precipitation is also in play in situations when the boundary layer smoke loading is less, as explored in Pennypacker et al. (2019), and have incorporated recognition of that relationship throughout. At the same time, we hypothesize a boundary layer semi-direct effect in which a boundary layer deepening is encouraged by the presence of boundary layer smoke. This would increase entrainment into the boundary layer and reduce cloud fraction. One could argue that a decrease in dropsize, from the presence of more smoke, could also support this entrainment. This effect doesn't contradict the existence of precipitating clouds removing aerosol, and the two effects can well be occurring, if not quite simultaneously. Models will be needed to attribute the different processes with confidence and more work is needed to ascertain precipitation susceptibility numbers. For now, the data have led us to the view that a radiatively-enhanced decoupling within the boundary layer can help explain the diurnal cycle in cloudiness that we document when more smoke is present in the boundary layer. Overall, we agree with the reviewers on the need for more nuanced language throughout our own document as well, and have paid attention throughout in our revised document.

The restructured manuscript contains an introduction that begins with a review of the current state of the research topic based on existing literatures, followed by several motivations of this particular study and clear statements of why the diurnal cycle becomes the focus of this study. The datasets, the compositing method, a monthly-mean overview of the BL thermodynamics and low-cloud regimes and amounts, and a time series description of the BL smoke loading and low-cloud features of the two August are provided in section 2. The main results of this study are presented in three parts: the observed low-cloud diurnal cycle as a function of BL smoke loading is shown in section 3, some explanations of the observed cloud diurnal cycle, both thermodynamical and microphysical, are presented in section 4, further influence on the altered diurnal cycle from the large-scale meteorological condition through entrainment are discussed in section 5. The last section 6 provides further interpretations of the observed features, summarizes the study, and emphasizes the importance of correctly resolving the low-cloud diurnal cycle towards establishing the regional radiative impact both in observational and model studies.

More specific responses are contained below, with the reviewer's comments provided in blue and our responses in black. Changes to the manuscript made in response to the reviewer are provided in italics.

Overall we thank the reviewers and editor for their time and energy towards improving this manuscript.

**REVIEWER 1:**

Throughout the manuscript, it is assumed that elevated BC concentrations cause certain changes in cloud cover, precipitation, temperature, BL structure etc. But in most (all?) cases the relation could very well be the opposite. For example, in Figure 7, it seems to me that the blue curves (in particular for 2016) show a diurnal variation in cloud fraction typical for a Sc deck while the red curves are more typical for a situation with Sc to Cu transition (cf. e.g. the review on stratocumulus by Wood, 2012). Could it not be that the overall large-scale meteorological conditions during certain periods are more favorable for Sc to Cu transition and afternoon Cu formation, i.e. that the subsidence is weaker? This would also favor a weaker inversion and more mixing of free tropospheric (polluted) air into the BL.

As is now more clearly indicated in Fig. 5 (now Fig. 4), the ``less" and ``more" smoky episodes tend to last multiple days. It seems unlikely that the transition would be preferentially sampled at the same location day after day. In addition, the deepening-warming concept for the transition needs to be refined for the situation common to August in the southeast Atlantic, given a heavy sunlight-absorbing aerosol loading. For example, we find that the days with the smokier boundary layers are characterized by more subsidence, rather than less. A decrease in the low cloud fraction with more subsidence, all else equal, is also independently documented in Adebiyi and Zuidema (2018). That said, the reviewer's comment encouraged us to consider the implication of the diurnal cycle for the transition more deeply. In response, we have added the following paragraph to the last section:

*"How does the diurnal cycle convolve with the stratocumulus-to-cumulus transition? The shift away from stratocumulus to more cumulus, when the boundary layer is smokier, might suggest that the transition is more likely to have started earlier (further upwind), under smokier conditions. The multi-day persistence of the time periods experiencing either enhanced or reduced boundary layer smoke loadings argues against the presence of a systematic sampling bias of the stratocumulus-to-cumulus transition within either composite. Nevertheless, the data analyzed in Fig. \ref{f12} support the inference that the presence of smoke within a non-overcast boundary layer may hasten the stratocumulus-to-cumulus transition through the increase in entrainment, if attributed here at least partially to radiation rather than only an accelerated entrainment from increased cloud droplet number concentrations \citep{Zhou17}. A contrasting observation occurs during the overcast days of 30-31 August, 2016, in which boundary layer smoke entrained far upstream \citep{Diamond18} supplied CCN concentrations of $\sim$800 cm$^{-3}$ at 0.4\% supersaturation. These are the only such days within the two Augusts in which the main stratocumulus deck reached Ascension. Although consistent with a delay in the transition by the aerosol-induced rain suppression \citep{Yamaguchi15}, a thoughtful attribution will also need to take large-scale conditions that begin to resemble those of September into account."*

Other examples:
o Page 7, lines 5-6: "A boundary-layer semi-direct effect is clear: when more smoke is present in the boundary layer, the diurnal-mean low-cloud cover decreases." To me, the relationship could very well work the other way.

We have removed this sentence as part of the overall rewrite.

o Page 7, lines 21-22: "The presence of more smoke reduces how frequently rain reaches the surface, although a slight late-night maximum and secondary late-afternoon maximum is still present." Rain could also decrease the smoke concentrations.

We have rewritten this section as shown below, acknowledging the opposite relationship in the last sentence:

*"The rain frequencies and median rain rates are less under smokier conditions, apparent even during times of day when the LWP increases (e.g., mid-morning during August, 2017). Cloud condensation nuclei concentrations (CCN) at 0.4\% supersaturation increase to daily-mean values of 790 cm\textsuperscript{-3} for the smokier conditions, compared to 114 cm\textsuperscript{-3} for the less smoky conditions. …. The higher LWPs that characterize the cleaner conditions will also help support rain production that reduces the accompanying aerosol loading \citep[e.g.,][]{Pennypacker19}.."*

o Page 8, lines 21-24: "Diurnally-resolved mean thermodynamic profiles indicate that the deepening of the smokier boundary layers occurs during the morning, followed by more significant shoaling during the night (Fig. 12a). The potential temperature of the smokier composite is the most warm near the surface in the mid-afternoon (14 LST), with the warmth moving upward in the late afternoon (17 LST)." What is the cause and the effect here?

We removed the second sentence, as we began to worry that perhaps island heating was also influencing this observation, and we also include the 400-600 m layer means in this figure now, to help address this concern. At this point in the manuscript, we are still just describing the observations, and not yet ascribing a cause and effect. We have simplified this section to make that more clear.

o Page 8-9, lines 30 and 1-2: "The warming is greater near the surface than higher in the boundary layer, which suggests that the absorbing aerosol may be preferentially located near the surface." Alternatively, the surface is just warmer in these cases.

We think this may be true. See previous response. We also address the issue of possible island heating within Section 2 describing the radiosonde data now.

Related to the comment above, I miss a discussion on the general BL structure in the region and its diurnal variation. What would be the typical BL variation (and largescale situation) for a day with thick stratocumulus clouds and a day with more stratocumulus above cumulus? It appears to me that in non-smoky conditions, the surface layer is stably stratified during night. In smoky conditions, it stays relatively well mixed (Fig 12). Could this indicate more mixing from above for the smoky conditions?

We have deepened our review of the diurnal cycle in the introduction. In addition, we discussed the general BL thermodynamical structures, degree of decoupling, and surface-observed cloud types in terms of monthly means in Section 2.4. The original Fig. 12, in which all of the profiles were averaged, was a bit misleading. The individual profiles indicate a decoupling that is typical regardless of the smoke loading, but the averaging lost that information. The decoupling is to be expected given the depth of the boundary layers. We have removed panels b-d to reduce this confusion.

In light of this reviewer's comment, we also included an additional analysis based on the WMO cloud code provided by the surface observers at the Ascension UKMO and embedded within the SYNOP provided every three hours. These show that during August 2016 and 2017, the dominant cloud type at over Ascension is "stratocumulus and cumulus with bases at different levels (CL=8)" (77%), the second most commonly-appearing cloud type is "cumulus with little vertical extent (CL=1)" (16%), and mid- and high-clouds only consist of 5% of the total occurrence (see below figure). "stratocumulus and cumulus with bases at different levels (CL=8)" attains its maximum occurrence at 9 am LST (~90% of the time), with a minimum occurrence at midnight (~65% of the time), while the diurnal cycle of "cumulus with little vertical extent (CL=1)" appear to be out-of-phase to CL=8, with a minimum at 9 am LST (~5%) and a broad maximum at midnight (~32%).

The original Fig. 5 has been modified to include this information, as well as to indicate which days fall into which composite. This reveals that there are only 2 days (August 30-31, 2016) that have smoky, overcast boundary layers, and all the other days are a combination of stratocumulus and cumulus. August 30-31, 2016, are discussed separately in several places throughout the modified manuscript, as they are interesting days that are different from the other days within the ``more smoky" composite. The large-scale conditions resemble those for September, with more subsidence and stronger free-tropospheric winds.

[Figure]

[Figure]

o Statistical significance, usage of "similar", "different", "clearly less" etc. and general uncertainties: There are in total 13 smoky days and 13 less-smoky days, how robust are the conclusions that you draw from this number of cases? Some examples:
o Page 7, lines 13-14: "The LWP is clearly less in the afternoon under smokier conditions in 2016, though not so in 2017." Compared to what? And is the difference robust/statistically significant? How many samples do you have in each bin?

A student's t-test shows that the differences in the mean LWPs every three hours between the two composites is statistically significant. That said, this isn't the best statistical test, as it assumes an underlying normal distribution, whereas the LWPs follow a highly-skewed positive-definite distribution. Instead, we have modified this figure to show the median and the 75th quartile, and discuss those qualitatively, which we feel communicates our point more clearly. Below is our revised text:

*"In both years, however, the median LWPs are less during much of the diurnal cycle when the boundary layer is smokier. The decrease is particularly noticeable at night, when the ``less" smoky days attain a nighttime LWP maximum that is matched by similar maxima in the nighttime rain frequencies and rates, matching expectations for stratocumulus decks, but the ``more" smoky days do not. Instead, the smokier days have the highest LWPs in the mid-morning after sunrise. This is evident in the upper quartile LWP values for both years."*

o Page 7, lines 14-15: "A more consistent feature between the two years is the reduction in LWP apparent after midnight in smokier conditions. Also consistent is an atypical increase in LWP that occurs mid-morning, after the cloud cover has begun to decline." Are these differences/variations significant?

See above

o Figures 8c&d and 9: what is the standard deviation for the different curves? Can you say with confidence that "Clouds are fewer and may be less vertically-developed in smokier boundary layers" and "… cloud tops are higher in the morning (06 – 12 LST) under smokier conditions…"?

Standard deviations are not as informative for quantities that don't follow normal distributions, such as rain frequencies, rain rates, and liquid water paths, and we have removed them from Fig. 8 (the new Fig. 9). The original Fig. 9 didn't include standard deviations. We just describe this figure in the text. That the cloud radar sees fewer clouds that are less vertically-developed, except during mid-morning, is consistent with the satellite-derived cloud top heights from a larger area shown in the original Fig. 10 (now Fig. 11) and with the boundary layer deepening evident in the soundings, so yes, we do think these are robust observations.

o Figure 13 and the discussion on page 9: what is the standard deviation? "The sub-cloud RH decreases during the day for higher smoke loadings…" but this occurs also for low smoke conditions? And is this decrease really robust? Same with "…with both peaking around 20 LST for light smoke loadings, and several hours later, around 2 am LST, for heavier smoke loadings." Is this really a robust variation?

The standard deviation in the original figure was that of the individual radiosondes about the three-hourly mean values. The number of measurements varied with each hour, as the 2016 measurements were 3-hourly and in 2017 they were 6-hourly (a detail that is clarified within the data section now). But we have removed these standard deviations, as they did not add new information and prefer the cleaner presentation lacking these.

The sub-cloud RH decrease we want to emphasize is the one in the afternoon. The decrease is replicated in higher cloud bases under smokier conditions, which is a completely independent, confirming observation. The two examples we show also indicate this clearly. So yes, we do think this is robust. We realize the sentence "The sub-cloud RH decreases during the day" is vaguely written, however, and have rewritten it as "The sub-cloud RH decreases during the afternoon". Regarding the nighttime increase, we are just reporting what we see. A new piece of information we have added is the 400-600m layer-mean diurnal cycle in the water vapor mixing ratio, which also shows a nighttime increase.

o Sections 5, 6 and 8: I am not sure these sections add much information. In general, I found them too speculative and did therefore not comment them in detail. In Figure 17, I would say that the differences are not significant.

The examples add value because they allow the analysis to move away from the composite means, which may be drawn to the extreme values, and because the examples show the vertical resolution in the radiosonde profiles that is missing in the composite means. We have integrated them into Section 3.

Section 6, on the top-of-atmosphere all-sky albedos, is now part of the discussion – it is useful as it demonstrates the possibility of a boundary layer semi-direct effect, which has not been acknowledged much for this region. Figure 17 (now Fig. 21) is reduced to just showing the probability distribution of the TOA albedos (i.e. the original inset plot). This acknowledges the reviewer's correct concern about our effort to break out the individual radiative impacts. We discuss the days represented within the distribution, and in particular the outlier: *"One day within the ``more" smoky composite, on 31 August 2016 (the outliner mentioned above), possesses both a high low-cloud fraction ($\sim$ 0.9) and an inordinately high albedo (near 0.4). On this day, cumulus coupling in the mid-morning strengthened a stratiform cloud that lasted through the day, and the SEVIRI-derived cloud droplet number concentration approached 250 cm$^{-3}$. The high all-sky albedo could be consistent with an aerosol indirect effect \citep[e.g.,][]{Lu18}, as ORACLES aircraft measurements to the southeast indicated little smoke above the boundary layer."* Although we don't report it, without the outlier, the difference between the two distributions is calculated to be significant at a 90% level through a student t-test.

We have removed Section 8 and the accompanying figure 14.

o Section 7: This section could be extended and written more carefully. For example:
o "Most fire emissions occur on the African plateau, ~1000 meters above sea level, with isentropic flow initially placing the biomass burning aerosols outflow above the low cloud deck." How often does this happen? And how would this influence your results/conclusions?

As detailed within Garstang et al., 1996, the continental mixed layer is about 3 km thick, meaning it reaches about 2 km above the African plateau. Their fig. 3 is reproduced below. The inversion heights they show are now corroborated by the radiosondes from Ascension. The temperature gradients in the free troposphere are fairly flat near the equator (see, e.g., Adebiyi et al., 2015). Isentropic flow would indeed place much of the aerosol above the low cloud deck. It still remains an open research question how and where the smoke enters the boundary layer exactly, and to do so for the days examined within this study is beyond what we can currently do. We have modified the sentence to include the word "much" to indicate that some may be advected directly into the boundary layer.

That said, the aerosol in the boundary layer that is sampled at Ascension must have been entrained from the free troposphere, as most of the aerosol advects off of the continent at approximately the same latitude as Ascension. If it went directly into the boundary layer, it would not reach Ascension. To make this clear, we have modified the text to include the following statement:

*"…Although most of the boundary layer aerosol at Ascension must have entrained from the free troposphere upwind of the island, to be detected at Ascension,"*

[Figure]

**Figure 3.** Annual subsidence (trade wind) inversion height (meters) for the South Atlantic Ocean [*von Ficker*, 1936] and the subcontinent. (Modified after *Preston-Whyte et al.*, [1977].)

o "Comprehensive visual inspection of the one-minute micropulse lidar extinction and depolarization profiles indicates that elevated smoke plume are frequent above Ascension in August." How often? This type of statistics would be highly relevant to show.

We have moved the discussion of the lidar to the last section and added the statement:

*"This study is not able to say much about the presence of smoke within the free troposphere above the low cloud deck accompanying either composite."*

Minor comments:

*Abstract,*

o Line 3: "... recent observations...", specify when.

o Line 13: ".... increase..." compared to what?

o Line 21: "...more easterly..." compared to what?

The abstract has been reconstructed to reflect the changes in the main text, and these points have been specified.

*Introduction,*

o I find the structure with unpublished figures in the introduction strange. I would recommend to include these figures (and the associated discussion) in Section 2 instead, as a part of the overview.

We have taken this point into consideration, and now only Fig. 1 is shown in the introduction just to highlight the frequent presence of smoke in the BL of Ascension Island during the biomass-burning season, especially August, which motivates this study, and this point has already been made by Zuidema et al. 2018 GRL.

o I miss some general question/questions stated at the end of the introduction.

Towards the end of the introduction, the main purpose of this study is more clearly stated.

o Line 6: "...stratocumulus thickening can occur..."

o Line 19: "This type of marine..."

Line 6 and line 19 have been corrected.

o Line 17: What do you mean with "The diurnal time scale is faster than that of meteorology..."?

This sentence is reworded as "*This takes advantage of the more numerous samples of the diurnal cycle, compared to of a particular weather or synoptic regime*"

*Data, overview and compositing approach*

o Page 4, line 5: What do you mean with "additional cloud"?

"Additional" is removed for clarification.

o Page 4, line 22: Define abbreviation (rBC).

rBC is defined here.

o Page 5, lines 3-4: What do you mean with "perhaps through their enclosure"?

This is clarified in the text by "perhaps from the building in which the radiosondes were kept."

o Page 5, line 23: How do you know that the retrieved LWPs include only Sc and shallow Cu?

Mostly because very little middle and high cloud was reported by surface observers during the two months, now shown in the new Fig. 3. The text is modified as "*Although technically an all-sky LWP, middle-and high-altitude clouds only occurred 5% of the time during the two August, with the radiometers not responsive to ice particles*."

o Page 6, line 14: "...indicate remarkably similar conditions..." Similar in terms of what? Cloud cover? rBC? CO? or all? And how do you define "similar"?

We have removed this sentence as part of the overall rewrite.

o Page 6, lines 16-17: "The exception is 21-23 August of 2017, when CO concentrations are close to the background value and rBC mass concentrations are..." What about 2016, 1-6 and 24-27?

August 1-6 and 24-27 from 2016 both contribute to the ``less smoky" composite, but August 21-23 of 2017 is the only time period when the rBC mass concentrations stayed within instrument sensitivity and CO concentrations decreased to a background level ~60 ppb, continuously.

o Page 6, line 19: "lasting for just a little shy of a week." Are you talking about 2016 or 2017?

This sentence is removed as part of the rewrite.

o Page 6, line 21: "...clearly reduces this time..." compared to what?

This sentence is rewritten as *"… tends to reduce during the more-smoky time periods …"*

*The cloud diurnal cycle as a function of the smoke loading*

o Page 7, lines 31-32: what do you mean with "...when they correlate more strongly with lower cloud fractions..."?

This sentence is reconstructed for clarification as *"… when the correlation between low-cloud fractions and cloud top heights is higher."*

*Explanations for the altered could diurnal cycle*

o Page 8, lines 5-6: "The radiosonde thermodynamic profiles (Fig. 11) indicate boundary layer depths that can exceed the cloud top heights documented in Fig. 10." But you don't see all cloud tops in Figure 10?

This sentence is removed to avoid confusion.

o Page 8, lines 7-8: "The deepening of the boundary layer is more well-defined..." compared to what?

Clarified, *"The deepening of the boundary layer is better-defined in August 2016 than in August 2017."*

o Page 8, line 28: I would suggest to change "fluxing" to "flux".

Corrected.

o Page 9, lines 5-6: Days with daily rBC variation less than 120 ng m-3 and 50 ng m-3, respectively, are selected from the "more" and "less" smoky days, and detrended. Please clarify.

o Page 10, line 10: "Under smokier conditions, the near-surface rBC values accumulate after mid-night.". This difference do not seem robust.

o Page 10, lines 12-13: "Overall, during a typical day, smoke settles towards the surface under smokier conditions, and is advected upwards when the surface layer is less smoky". This formulation is unclear to me.

This subsection is now removed from the manuscript for the following reasons: a. the detrending approach raised concerns from the reviewers comparing to a regular departure from the daily-mean method, b. the interpretations we provided based on the composites were too speculative without sufficient amount of evidences to support them, c. the daily variability of the smoke loading is high, leaving the analysis less robust, and d. too many factors are contributing to the BL loading of rBC, such as entrainment from the free troposphere (mixing from above), horizontal advection, and of course, the

precipitation removal process through coalescence scavenging, and in addition, these processes are happening at different time scales, making a synthetic statement on the general diurnal cycle of smoke loading at Ascension extremely hard. Therefore, we choose not to include this analysis in the revised manuscript.

*Examples*

Note this part no longer a subsection and these examples and the corresponding discussions are now included in last part of section 3.

o Page 10, lines 20-21: "…indicate cumulus clusters in the morning…" or cumulus under stratocumulus?

We reconstructed this part, and the use of "cumulus clusters" is refrained as it is causing confusion. Instead, we write *"Precipitation-producing cumulus surmount the full boundary layer on both mornings, with smaller cumuli dispersed throughout the day and night."*

o Page 10, line 22: What do you mean with "upper level cloud"?

We get rid of "upper level cloud" in the text and use "stratiform cloud" instead.

o Figure 15: what is RF in the figure? And what does the KAZR radar mainly see? I assume it is mostly precip-sized particles, this should be clarified.

RF represents rain frequency, and this is defined in the caption. The data section now includes the following sentence: *"The KAZR has a sensitivity of -29 dBZ at 2km at a range resolution of 50 m. The sensitivity and resolution is enough to detect most but not all thin, non-precipitating clouds, and ceilometer data are also invoked to improve detection of all clouds."*

o Page 10, line 23: "…with only scattered cumuliform clouds remaining…" How do you see this? And are you talking about both cases?

Rewritten as: "*By the mid-afternoon, the stratiform cloud has thinned, with scattered cumuliform clouds dominating the radar and satellite imagery.*"

**REVIEWER 2:**

The larger concerns of Reviewer 2 are an overemphasis on the casual link from smoke to clouds/precip, and, the aerosol forcing analysis surrounding Fig. 17. We have taken care to describe the smoke/precipitation relationship more agnostically (likely both pathways smoke->precipitation suppression and precipitation->aerosol washout are active, until the models are able to reproduce the observed liquid water paths it will remain controversial which pathway is dominant; the smoke->smaller droplets->more entrainment will also need to be considered).

The aerosol forcing analysis and discussion is now included in the last section of the revised manuscript. The main point of this is to indicate a boundary layer semi-direct effect may also contribute, increasing the number of myriad aerosol-cloud interactions that as a discussion to solely stimulate further research interests on this. We are not aiming to resolve an aerosol forcing estimation based on our available

datasets. The reviewer raises a good point that an increase in LWP for the same cloud fraction within the smokier composite could also explain the increase in albedo, rather than the change in aerosol. We looked at this using the LASIC LWPs, and found a range of LWPs. The correct way to do this would be to evaluate the satellite-derived LWPs for the same domain, if we are going to compare across the entire domain. All told, we recognize the analysis shown in Fig. 21 is premature and, we refrain ourselves from trying to break out the individual radiative impacts in this manuscript. Instead, we decided to just show the histogram (originally the inset plot) as Figure 21 in the revised manuscript to report the overall TOA all-sky albedo changes as a function of BL smoke loading, serving as a guide and a stimulation for future work in this region on this topic.

p1 l15 "direct aerosol radiation" is probably better as "increase in reflected solar radiation in clear sky attributable to aerosol" or so?

The abstract no longer includes mention of the radiative analysis we have done.

p2 l3 "decks"

p2 l17 "prior"

Corrected.

p3 l17 The authors should define what they mean by "meteorology" here.

This has been rewritten as "*This takes advantage of the more numerous samples of the diurnal cycle, compared to of a particular weather or synoptic regime*"

p4 l8 In fact the airport is shown in plain

The legend of the original Fig. 4 (now Fig. 2) says that the airport distribution is shown in solid lines, as oppose to the AMF1 site which is shown in dashed lines. In the revised manuscript we no longer mention this detail within the main text to be consistent with the description of the other figures.

p5 l3 Here and later, "meter" should be abbreviated as "m". Same later for "seconds" → "s"

Corrected, here and after.

p5 l22-23 This sentence I do not understand. One radiometer failed in 2016, and another one was used in 2017? Where is the problem?

We meant 2016, sorry for the confusion, MWRP retrievals is used for 2016, and MWR retrievals is used for 2017, and these are carefully put in the revised manuscript as *"The August 2016 (2017) LWPs are physically retrieved from the microwave radiometer profiler (two-channel microwave radiometer) using the standard ARM operational retrieval algorithm (Turner et al., 2007)"*.

p6 l3 The definition "liquid-only water+suspected water" requires slightly more explanation. Is "water" = "liquid cloud"? Or rather open ocean surface?

This is rewritten as "*liquid-water cloud and suspected liquid-water cloud*"

p6 l10 The difference in the two MODIS retrievals (the one included in the CERES product and the other one used) should be explained, ideally by providing the references for the two retrieval algorithms.

References are added for CERES products and MODIS level-3 dataset in the revised manuscript.

p6 l12 Is the difference between MODIS and SEVIRI due to the fact that SEVIRI re- solves the diurnal cycle? i.e. is SEVIRI = MODIS if SEVIRI is sampled at the (four? or more?) MODIS overpass times?

Diurnal cycle could be one of the reasons for the difference between MODIS and SEVIRI. The two satellite don't quite sample at the same time, and are also different in their viewing zenith angles and pixel sizes, as well as retrieval algorithms. We don't quite know the full cause for the differences and are just reporting them. More discussion is added to the revised manuscript.

p6 l16 What is the CO concentration background value?

50-60 ppb in the clean Southern Ocean, according to Allen et al. 2011 and Shank et al. 2012. This is added to the main text in the revised manuscript.

p6 l22 Another obvious explanation is the opposite causality: smoke is low when it is washed out by rain, and larger when there is no rain

In the revised manuscript, we now just report that the precipitation is observed to be less frequent and less intense during smoke events and choose not to discuss possible causes here in this subsection. The new sentence is "*Overall, regardless of cloud type, precipitation is less frequent and less intense during smokier time periods*"

p6 l28 What motivates the current choice of thresholds, rather than the actual terciles?

The choice of these thresholds is based on rounding off the terciles for the ease of readership without resulting in any difference in the actual smokier/cleaner day selection. This clarification is added to the text in the revised manuscript. The sample sizes are the same and the days that contribute to each composite are provided.

p7 l6 "is clear" - there is no proof for causality here. Why not a more cautious language just stating the co-variation?

This section has been reconstructed with more cautious language just stating the observed co-varying features without suggesting any causality, and interpretations and explanations to these observations are introduced in Section 4 in the revised manuscript. The sentence in question has been rewritten as "*In both years the low cloud fraction is reduced during all hours of the day during the smokier time periods, with a more pronounced late-afternoon cloud reduction during the smokier time periods in August, 2016*."

p7 l11 "more to a measure of cloud thickness than cloud fraction" - is this in fact all- sky LWP? Because else the sentence is mis-leading: it only is cloud thickness, and not cloud fraction (even if of course ancillary knowledge is that the two are usually correlated).

We now carefully call this "cloud liquid water path" to make it clear to the reader that clear sky samples (i.e. LWP = 0) are removed from the composite, and have removed mention of cloud thickness. Indeed, given that the boundary layer clouds are often separated between two layers, the term 'cloud thickness' isn't quite right.

p7 l21 The causality presumably is the inverse: there is less smoke when it rains

We have rewritten this sentence to just report the observations: "*The decrease is particularly noticeable at night, when the ``less'' smoky days attain a nighttime LWP maximum with corresponding maxima in the nighttime rain frequencies and rates, but the ``more'' smoky days do not. Instead, the smokier days have the highest LWPs in the mid-morning after sunrise. This is evident in the upper quartile LWP values for both years*." In the next paragraph we mention that the daily-mean CCN concentration at 0.4% SS increase from 114 cm$^{-3}$ to 790 cm$^{-3}$ in the smokier time periods. The possible causality suggested by the reviewer is added to the revised manuscript as "*The higher LWPs that characterize the cleaner conditions will also help support rain production that reduces the accompanying aerosol loading \citep[e.g.,][]{Pennypacker19}.*" We also note in the manuscript though *"… that the post-sunrise increase in the upper quartile of LWP is not accompanied with an increase in rain*…"

p11 l26 Also LWP differences play a – presumably dominant – role.

This is a good point. We decided to just show the histogram (originally the inset plot) as Figure 21 in the revised manuscript to report the overall TOA all-sky albedo changes as a function of BL smoke loading, serving as a guide and a stimulation for future work in this region on this topic.

p12 l2 So far I would have understood the topic is cloud adjustments to aerosol- radiation interactions ("semi-direct effect"), not so much aerosol-cloud interactions (what was called "microphysical" earlier in this manuscript).

The whole paragraph is reconstructed, and this sentence is removed. See responses above.

p14 l21 Diabatic warming, not necessarily radiative: precipitation can play a role (and presumably usually does, since mostly radiation acts to cool the layer)

This sentence just explores how much of a vertical ascent could be attributed to shortwave absorption. Later sentences mention the additional contribution from latent heating. To clarify we have restated 'diabatic radiative warming' as 'shortwave absorption'.

p24 Fig. 3: The median obviously is also shown. (same p25 Fig 4)

The caption has been corrected.

p28 Fig. 7: presumably the grey shading indicates sun below horizon?

Yes, and this is stated in the captions.

p35 Fig. 14: slightly more explanation is necessary on how the detrending is done, and on what exactly is the "Delta" value on the y-axis. Why not a deviation from the mean?

This figure and the corresponding subsection have been removed from the manuscript.

**REVIEWER 3:**

The reviewer's main concerns were to better separate the review and motivation within the introduction, clarify the relevance of the diurnal cycle, and structure the manuscript into sections of more equal length. We have followed up on these suggestions, and believe have constructed a stronger piece of writing that is difficult to communicate within this response other than to provide the revised manuscript. Responses to the specific comments are provided below.

1. page 1, line 0 (henceforth 1-0 etc.): In the title, it is not clear what "the diurnal cycle" refers to (clouds, obviously, from the text). Also, it is not clear that the link between smoke and clouds is at the center of this study. Suggestion "Smoke impacts on amount and diurnal cycle of low marine boundary layer clouds in the SE Atlantic" – or something similar.

The title is now changed to *"The diurnal cycle of the smoky marine boundary layer observed during August in the remote southeast Atlantic".*

2. 1-4: awkward sentence. Smale is not only "highly absorbing" in August. Please try rewording.

This sentence has been removed as part of a larger rewrite.

3. 1-9: What is a "surface-based" mixed layer?

This phrase is changed to the "sub-cloud layer".

4. 1-11: I don't understand what is meant by "increasing the moisture stratification" – how can a stratification increase?

This reworded for clarification, as "*A discrete moisture stratification establishes itself then between the cloud and sub-cloud layer…*".

5. 1-15: This sentence seems confusing and counter-intuitive to me. Absorbing aerosol decreases radiation at TOA, correct? So taking the difference between the least and most smoky terciles will yield a negative figure, i.e. a reduction. I suggest presenting this fact in this way.

We have removed discussion of this analysis from the abstract

6. 4-4: The subsection beginning here seems a mixture of process discussion ("oro- graphic lifting cleary induces...") and a description of the context of the measurements. I suggest moving the former to the state-of-the-art section, and giving the remainder a sub-title (2.1 Context of measurements, or similar)

The beginning paragraph of this section sets the context and rationalization of the choice of these datasets, and therefore, should leads the following subsections. We would like to maintain this structure, as it contains too much detail to be in the introduction.

7. 6-3: "suspected water" – I assume this refers to liquid water?

Yes, and "liquid" has been added for clarification.

8. 6-5: apparent LIQUID-water clouds (?)

"liquid" is added.

9. 6-7: How is this evaluated?

This is further discussed when we present the data in the discussion and summary section (Section 6). Here, in Section 2.2, we just want to introduce the data we are going to show. Another note is that this TOA radiation budget assessment has been reduced in the revised manuscript, which no longer evaluates the relationship between CERES TOA all-sky albedos and CERES low-cloud fractions, but rather a report of the observed changes in TOA all-sky albedo as a function of the BL smoke loading, serving as a guide and a stimulation for future work in this region on this topic.

10. 6-9: "For context" – what does this mean?

This phrase is removed.

11. 7-3: sunrise and sunset are not indicated in figure 7.

The gray shading in each plot represents nighttime, and therefore, sunrise and sunset is the intersections between daytime and nighttime. This is mentioned in the captions.

12. 7-3: The peak is barely visible, therefore the reader cannot be sure that this 'peak' actually represents all or most individual days. One solution to this might be to show deviation from the mean cloud fraction of each day (daily cloud fraction anomaly) on the vertical axis instead of the actual cloud fraction.
13. 7-9: See comment on cloud fraction, line 7-3.

We'd rather keep the absolute values of the low cloud fraction on these plots. However, the reviewer did raise a good point, and for that reason, we added the diurnal means and diurnal magnitudes into the text.

14. 8-13: What is a 'surface-based' layer?

The use of this "surface-based" phrase is removed from the manuscript, and we use the term sub-cloud layer throughout instead, which is well-accepted within the literature.

15. Fig. 21: I very much like this conceptual summary of your work. However, I ask you to consider the following points: Within each of the four phases, can the blue and red clouds be placed next to each other for visual clarity? 1) In the morning I find it hard to make out the exact borders of the blue clouds. 2) blue is in front of red in the first panel, behind in all others. 3) placing 'mignight', 'noon' and 'midnight' on the horizontal axis might help the reader. 4) the warming of the BL progresses during daytime. The red horizontal arrow in the morning should therefore point to the right. 5) Please explain all abbreviations in the caption.

We really appreciated the reviewer's suggestions for improving the quality of this schematic plot. All suggestions are incorporated into the revised figure.

We thank the reviewer for pointing out the technical details at the end of the review, and we have incorporated the corrections.

---

## Referee Report (RR1)

Second review of "Low cloud reduction within the smoky marine boundary layer and the diurnal cycle" by J. Zhang and P. Zuidema

The authors have done a thorough revision of the manuscript, it has improved substantially, both in terms of contents and structure. The results are interesting and relevant for the scientific community and they are now analyzed and presented in a more objective way. However, there are still many minor issues that I think need consideration, as outlined in my comments below.

Main comments:
- In general, I find that the manuscript could be more concise, both in terms of text and figures; 22 figures in a manuscript seems a bit excessive. Could the different figures showing radiosonde data (for example) be combined? And could some figures that are shown more as support for arguments be moved to a supplementary section?
- Considering my many minor comments below (and I did not include all of them), I think the manuscript would benefit from another thorough read-through to check that all sentences are clear.
- Discussion regarding smoke transport: I agree with the authors that the smoke transport most likely occurs (primarily) above the boundary layer. But I don't think that this can be taken for granted, and I think this uncertainty should be more clearly reflected on/considered in Section 5, in particular in the discussion related to the smoke loading and the strength of the inversion.
  - The smoke measurements are at the surface, and (unfortunately), we don't know if there is more or less smoke above –the vertical structure could maybe explain some of the relation between inversion strength and smoke loading. On lines 13-19, you discuss a case with high smoke in the BL, but advection of air with smoke in the free troposphere, which is quite confusing.
  - There are two lidar profiles shown, but the purpose of these are not clear to me. I don't think they add much information as they are only for a single case (and it's not clear if this is a typical case). Furthermore, I don't think the lidar profiles can be trusted in the boundary layer, so they cannot help understanding the relation between the surface concentrations and the free troposphere.
  - Line 10: from where do you get that the warming is only 0.5K?
  - Wind pattern in the free troposphere: it is not clear to me from Figure 20 that "easterlies and north-easterly winds become more frequent when the BL is smokier". "Winds are stronger in the BL and lower FT when the smoke loading is high…", but this is not true for 2016?

Minor comments :
*Abstract:*
- Line 8: "… decreases further." Further than what?
- Lines 11-14: The sentence starting with "A reforming…" and "After the sun rises…": I assume that what you describe here is for smoky conditions? It needs to be clarified that during smoky conditions you do not (always) reform the stratiform layer and that you therefore have less chance of recoupling.
- Line 16: I would suggest changing "reestablishing" to "strengthening" as the cumulus convection only locally and temporarily couples the surface, sub-cloud and cloud layer.

*1. Introduction*
- Page 2, line 34: I think "maxima" should be "maximum"?

*2. Data, overview and compositing approach*
- Page. 7, lines 8-10: The sentences starting with "Pennypacker et al…." and "This suggests" need clarification.
- Page 7, line 21: "The monthly-mean profiles…" Are you referring to Figure 2 or Figure 6 here..? I assume Figure 6, but still it's not clear to me how you see that there is a decoupling. Perhaps it could also be good to include the layers in the figure?
- Page 8 lines 4-5: I think it should be "ones" instead of "one" as you are referring to two events in mid-August and two events at the end of August, one for each year.
- Page 8, line 11: I suggest changing "reduce to" to "approach".
- Page 8, line 13: I suggest changing "observed reports" to "observations".

*3. The cloud diurnal cycle as a function of the smoke loading*
- Page 9, line 2: I suggest changing "not completely similar" to "different" or "slightly different".
- Page 9, line 3: I suggest changing "…a nighttime LWP maximum has corresponding…" to "there is a nighttime LWP maximum that corresponds with.."
- Page 9, lines 4-5: The sentence starting with "A secondary maximum…." is vague. I suggest reformulating or remove.
- Page 9, lines 19-23: Are these characteristics specific for smoky conditions? This is not clear.
- Page 9, lines 25-29: Please check sentence structure and make it clearer why there is less (local) coupling with less cumulus.
- Page 9: lines 29-30: Why are these two cases selected? I think this needs to be at least briefly motivated.
- Page 9, line 30: I don't think "possesses" is the right word here. Perhaps "display"?
- Page 9, line 33: How do you know that the stratiform layers come primarily from detrainment..?
- Page 10, lines 5-6: Please explain how you see that the sounding profiles are decoupled all day (i.e. exactly which sub-figure(s) are you referring to).

*4. Explanations for the altered cloud diurnal cycle*
- Page 10, lines12 and 13: I would suggest changing "can" to "could".
- Page 10, line 22: Is "capped" really the right word? I would think that this refers to something *above* the layer?
- Page 10, line 31: I would suggest changing "better-defined" to "more defined".
- Page 11, line 6: I would suggest changing "if more" to "although".
- Page 11, line 6: I would suggest changing "with the hint of" to "indicating a possible".
- Page 11, line 9: I would suggest changing "an afternoon" to "the afternoon".
- Page 11, lines 15-17: Are you referring to the non-smoky BL here? The description does not fit so well with the smoky BL.
- Page 11, lines 17-18: Please check sentence structure.
- Page 11, line 25: When you mention "cumulus coupling", I think it would be good to make clear that this is a local, intermittent coupling.

*6. Discussion and summary*
- Page 12, line 13: I suggest changing "its coupling to" to "potentially coupling it to".
- Page 13, line 10: I suggest changing "discourages" to "inhibiting".
- Page 13, line 23: "Why is cumulus-coupling…", is this statement referring to smoky conditions?

- Page 13, lines 7-9: Please check the sentence structure.
- Page 13, line 15: "Meanwhile, 800 hPa winds are more easterly/northeasterly….". As mentioned above, I don't think this is really clear from figure 20.
- Page 13, lines 25-26: Please check sentence structure.

*Figures*
- Figure 9c and d: Are these for 2016 or 2017 or both?

---

## Author Response (AR3)

**Responses to the second review of "Low cloud reduction within the smoky marine boundary layer and the diurnal cycle" by J. Zhang and P. Zuidema**

*\*We thank the editor for pointing out a mistake in Fig. 18, and it has been corrected in the newer version of the manuscript.*

We would like to thank the reviewer for his/her thorough read-through of the revised manuscript, as well as the thoughtful suggestions helped us improve the quality of this manuscript.

The revised manuscript now contains 18 figures, with four figures moved to a supplement file, which helps the manuscript become more concise. Careful read-throughs are done to make sure the sentence structures and wording are clear and easy to understand.

More specific responses are contained below, with the reviewer's comments provided in blue and our responses in black. Changes to the manuscript made in response to the reviewer are provided in italics.

Again, we thank the reviewers and editor for their time and energy towards improving this manuscript.

**Main comments:**

- In general, I find that the manuscript could be more concise, both in terms of text and figures; 22 figures in a manuscript seems a bit excessive. Could the different figures showing radiosonde data (for example) be combined? And could some figures that are shown more as support for arguments be moved to a supplementary section?

We found it hard to combine/merge the figures containing the radiosonde data as they are shown in different parts of the manuscript and combining any of them will impair the order of arguments we are trying to make. Instead, we have moved the original Figs. 6, 7, 19 and 21 to a supplement file. These mostly provide context for other, more key figures.

- Considering my many minor comments below (and I did not include all of them), I think the manuscript would benefit from another thorough read-through to check that all sentences are clear.

The revised manuscript has taken all the minor comments into account and revised accordingly. Several careful read-throughs by both authors are done to make sure all sentences are clear and as concise as possible.

- Discussion regarding smoke transport: I agree with the authors that the smoke transport most likely occurs (primarily) above the boundary layer. But I don't think that this can be taken for granted, and I think this uncertainty should be more clearly reflected on/considered in Section 5, in particular in the discussion related to the smoke loading and the strength of the inversion.

Section 5 has been reworked to reflect how we think the FT smoke could affect our interpretation of the inversion strength analysis. "*More work is needed to fully ascertain how the boundary layer aerosol at Ascension arrived there*" clearly states that we acknowledge possible various pathways of the transportation of the BL smoke at Ascension, and thus, a FT smoke transport has not been taken for granted. We have also moved our discussion of the lidar profiles to this section, away from Section 6. This at least indicates that aerosol is often present in the free troposphere as well.

○ The smoke measurements are at the surface, and (unfortunately), we don't know if there is more or less smoke above –the vertical structure could maybe explain some of the relation between inversion strength and smoke loading. On lines 13-19, you discuss a case with high smoke in the BL, but advection of air with smoke in the free troposphere, which is quite confusing.

We have rewritten this section to include the discussion of the more comprehensive lidar analysis, and expanded the discussion of the case slightly to be more communicative with the readership.

○ There are two lidar profiles shown, but the purpose of these are not clear to me. I don't think they add much information as they are only for a single case (and it's not clear if this is a typical case). Furthermore, I don't think the lidar profiles can be trusted in the boundary layer, so they cannot help understanding the relation between the surface concentrations and the free troposphere.

The purpose of showing the lidar extinction profiles on top of the potential temperature profiles is to show that the same winds transport both the above-cloud aerosol layer and cooler temperatures, compared to the mean free-tropospheric temperature profile. That said, we agree that this is a single example which does not guarantee a general colocation of FT smoke layer and cooler temperatures. Therefore, we have left the figure out of the main text, and put it into a supplement file as a support for this argument instead. We also agree that the lidar extinction profiles are not quantitatively trustworthy in the BL, and the purpose of these extinction profiles, like mentioned above, was to highlight, qualitatively, the existence of a FT smoke layer. The limitations to the lidar retrievals are emphasized more within the manuscript.

○ Line 10: from where do you get that the warming is only 0.5K?

The smoke-induced warming is estimated by taking the difference between the blue (less smoky) and red (smokier) curves in Fig. 17 a (see the green arrows below). This is true for both the 200 – 400 m (solid) and 400 – 600 m (dashed) layer-averaged potential temperature.

[Figure]

○ Wind pattern in the free troposphere: it is not clear to me from Figure 20 that "easterlies and north-easterly winds become more frequent when the BL is smokier". "Winds are stronger in the BL and lower FT when the smoke loading is high...", but this is not true for 2016?

We see the reviewer's point. The frequency of easterlies at inversion tops (~800 hPa or ~2 km) is not that different between the smokier and less smoky conditions. However, it is true that the 2 km northeasterlies are more frequent when the BL is smokier. We have modified our statement to "*A more comprehensive assessment of the winds, shown separately for the two years as a function of smoke loading (Fig.~\ref{f17}), indicates that at 2 km, just above the inversion top, times with stronger north-easterly winds become more frequent when the boundary layer is smokier (Fig.~\ref{f17}).*" Winds are stronger in the FT right above the inversion tops both in 2016 and 2017 when the BL smoke loading is high, whereas in the BL, wind speeds are clearly higher in 2017, but not that different (slightly higher) in 2016.

**Minor comments:**

Abstract:

- Line 8: "... decreases further." Further than what?

This is rewritten as *"The afternoon low-cloud minimum is more pronounced on days with smokier boundary layer."*

- Lines 11-14: The sentence starting with "A reforming..." and "After the sun rises...": I assume that what you describe here is for smoky conditions? It needs to be clarified that during smoky conditions you do not (always) reform the stratiform layer and that you therefore have less chance of recoupling.

Yes, we are describing the smokier conditions here. To avoid confusion, this part is rewritten as *"Under these conditions, the nighttime stratiform cloud layer does not always recouple to the sub-cloud layer, and the decoupling maintains more moisture within the sub-cloud layer. After the sun rises, enhanced shortwave absorption in a smokier boundary layer can drive a vertical ascent that…"*

- Line 16: I would suggest changing "reestablishing" to "strengthening" as the cumulus convection only locally and temporarily couples the surface, sub-cloud and cloud layer.

Suggestion taken, reworded as so.

1. Introduction

- Page 2, line 34: I think "maxima" should be "maximum"?

Corrected.

2. Data, overview and compositing approach

- Page. 7, lines 8-10: The sentences starting with "Pennypacker et al...." and "This suggests" need clarification.

This part is rewritten as "*The ``less" smoky days will still contain some smoke in the boundary layer. \cite{Pennypacker19} document that low-aerosol days at Ascension are dominated by precipitation scavenging, similar to the Azores \citep{Wood17}. Their analysis, which includes twelve days from August, 2016, indicates the ``less" smoky days can capture the dominant features of the low-aerosol cloud diurnal cycle.*"

 Are you referring to Figure 2 or Figure 6 here..? I assume Figure 6, but still it's not clear to me how you see that there is a decoupling. Perhaps it could also be good to include the layers in the figure?

Yes, we are referring to Figure 6 here, and we have added a reference to the figure at the end of this sentence. This sentence is rewritten as "*The monthly-mean profiles indicate a stratification in the water vapor mixing ratio profiles that imply a decoupling coinciding with the sub-cloud transition layer (Fig. S2 b and e).*"

The decoupling is seen through the stratification in the moisture (water vapor mixing ratio) profiles between the upper and lower boundary layer (see the green boxes below). We think this stratification in the moisture profiles is rather a clear indication of a decoupled boundary layer, and we would like to keep the figure as it is now without adding more lines to complicate this plot. In light of the reviewer's comment on making the manuscript more concise, figure-wise, we decided to move this figure to a supplement file since it is not one of the figures that our key arguments are based on.

[Figure]

- Page 8 lines 4-5: I think it should be "ones" instead of "one" as you are referring to two events in mid-August and two events at the end of August, one for each year.

Corrected.

- Page 8, line 11: I suggest changing "reduce to" to "approach".

Actually line 8, changed to "*approach*".

- Page 8, line 13: I suggest changing "observed reports" to "observations".

Changed to "*observations*".

3. The cloud diurnal cycle as a function of the smoke loading

- Page 9, line 2: I suggest changing "not completely similar" to "different" or "slightly different".

Changed to "*slightly different*".

- Page 9, line 3: I suggest changing "...a nighttime LWP maximum has corresponding..." to "there is a nighttime LWP maximum that corresponds with…"

Suggestion taken, rewritten accordingly.

- Page 9, lines 4-5: The sentence starting with "A secondary maximum…." is vague. I suggest reformulating or remove.

This sentence is removed from the revised manuscript.

- Page 9, lines 19-23: Are these characteristics specific for smoky conditions? This is not clear.

Yes, they are, except for the 08 LST samples. These sentences are rewritten as "*Cloud top heights in a smokier boundary layer are approximately one km at the lowest cloud fractions, indicating that the upper stratiform cloud layer is likely not present. The exception is in the morning just after sunrise. Almost all 08 LST samples (Fig. \ref{f9}, green crosses), regardless of smoke loading, have a cloud effective height higher than 1.5 km, consistent with the radar observations (Fig. \ref{f8}).*" for clarification.

- Page 9, lines 25-29: Please check sentence structure and make it clearer why there is less (local) coupling with less cumulus.

This part is rewritten as "*The shift between the two cloud types is most pronounced from midnight until 9 LST (not shown), suggesting the upper-level stratiform layer has more trouble reforming at night under smokier conditions.*"

- Page 9: lines 29-30: Why are these two cases selected? I think this needs to be at least briefly motivated.

In the main text, we have slightly rewritten this as "*Two days, chosen because MODIS satellite visible imagery are available for both overpasses (Fig.~\ref{f11}), also depict meaningful features in their sounding profiles that may otherwise be averaged over within composite means,*" to be more clear.

- Page 9, line 30: I don't think "possesses" is the right word here. Perhaps "display"?

This sentence is rewritten as "*One day, 12 August, 2017, is included in the ``more'' smoky composite (daily-mean rBC of 663 ng m\textsuperscript{-3}), while the smoke loading on 20 August, 2017, was intermediate at 346 ng m\textsuperscript{-3}.*"

- Page 9, line 33: How do you know that the stratiform layers come primarily from detrainment…?

We agree that this is a subjective determination based on the radar and satellite imageries. We have reworded the sentence as "*Thin stratiform layers, at times detected by the ceilometer but not by the cloud radar, primarily appear associated with detrainment from cumulus in the satellite imagery, although can also be detached.*"

- Page 10, lines 5-6: Please explain how you see that the sounding profiles are decoupled all day (i.e. exactly which sub-figure(s) are you referring to).

The decoupled boundary layer is indicated primarily by the stratification in the moisture (water vapor mixing ratio) profiles, however, one can also easily tell whether the BL is well-mixed (coupled) or decoupled through potential temperature and equivalent potential temperature profiles. We have added a reference to the exact sub-figures that indicate the decoupling feature.

This is rewritten as "*The water vapor mixing ratio ($q_v$) profiles suggest that the boundary layer is decoupled all day, but the moisture stratification is more clear at night…*"

4. Explanations for the altered cloud diurnal cycle

- Page 10, lines12 and 13: I would suggest changing "can" to "could".

Suggestion taken.

- Page 10, line 22: Is "capped" really the right word? I would think that this refers to something *above* the layer?

We rewrote the sentence as "*The sub-cloud layer is typically well-mixed, with a mean $\theta$ of $\sim$296 K and mean $q_v$ of $\sim$13 g kg\textsuperscript{-1}.*"

- Page 10, line 31: I would suggest changing "better-defined" to "more defined".

Suggestion taken.

- Page 11, line 6: I would suggest changing "if more" to "although".

Suggestion taken.

- Page 11, line 6: I would suggest changing "with the hint of" to "indicating a possible".

This sentence is rewritten as "*…with a noontime static instability relative to the lower layer that may reflect island heating.*"

- Page 11, line 9: I would suggest changing "an afternoon" to "the afternoon".

Suggestion taken.

- Page 11, lines 15-17: Are you referring to the non-smoky BL here? The description does not fit so well with the smoky BL.

We made a mistake in this sentence which causes the confusion, and this is rewritten as "*The diurnal cycle in the sub-cloud $rh$ (Fig.~\ref{f15}c) reflects characteristics of both the temperature and $q_v$ diurnal cycles, with a shift in the diurnal minimum from midday to mid-afternoon as the smoke loading increases.*"

- Page 11, lines 17-18: Please check sentence structure.

This sentence is removed from the revised manuscript.

- Page 11, line 25: When you mention "cumulus coupling", I think it would be good to make clear that this is a local, intermittent coupling.

This sentence is removed from the revised manuscript to avoid confusion. In addition, we have also made sure that "*intermittent*" is included when "*cumulus coupling*" is mentioned.

6. Discussion and summary

- Page 12, line 13: I suggest changing "its coupling to" to "potentially coupling it to".

This part is rewritten as "…*helping to couple the cloud to its surface moisture source…*"

- Page 13, line 10: I suggest changing "discourages" to "inhibiting".

Suggestion taken, changed to "*inhibits.*"

- Page 13, line 23: "Why is cumulus-coupling…", is this statement referring to smoky conditions?

Yes, "*under smokier conditions*" has been added to make the statement clearer.

- Page 13, lines 7-9: Please check the sentence structure.

We assumed the reviewer meant Page 14, lines 7-9, which is rewritten as "*Although still a decoupled boundary layer, cumulus coupling in the mid-morning of 31 August strengthened a stratiform cloud that lasted through the day…*"

- Page 13, line 15: "Meanwhile, 800 hPa winds are more easterly/northeasterly….". As mentioned above, I don't think this is really clear from figure 20.

We realized that only north-easterlies at 800 hPa are more frequent under smokier conditions. This part is modified accordingly as in revised section 5. In the revised manuscript, it says, "*Meanwhile, north-easterlies at 800 hPa are more frequent (Fig. \ref{f17} d and h), favoring lower-level transport of smoke off of the African continent that will reside closer to the stratocumulus cloud tops.*"

- Page 13, lines 25-26: Please check sentence structure.

We assumed the reviewer meant Page 14, lines 25-26, which is rewritten as "*Overall the top-of-atmosphere all-sky albedos decrease (Fig. S4, the difference is significant at the 90\% level) as the smoke loading in the boundary layer increases, with the exception of 31 August, 2016.*"

Figures

- Figure 9c and d: Are these for 2016 or 2017 or both?

These are for both 2016 and 2017. Figure caption is edited to make this information clearer.

[revised manuscript text omitted]